# Temporal perturbation of ERK dynamics reveals network architecture of FGF2/MAPK signaling

Yannick Blum[1,†], Jan Mikelson[2,†], Maciej Dobrzyński[1,†] , Hyunryul Ryu[3,‡], Marc-Antoine Jacques[1], Noo Li Jeon[3], Mustafa Khammash[2] & Olivier Pertz[1,*]

## Abstract

**Stimulation of PC-12 cells with epidermal (EGF) versus nerve (NGF) growth factors (GFs) biases the distribution between transient and sustained single-cell ERK activity states, and between proliferation and differentiation fates within a cell population. We report that fibroblast GF (FGF2) evokes a distinct behavior that consists of a gradually changing population distribution of transient/sustained ERK signaling states in response to increasing inputs in a dose response. Temporally controlled GF perturbations of MAPK signaling dynamics applied using microfluidics reveal that this wider mix of ERK states emerges through the combination of an intracellular feedback, and competition of FGF2 binding to FGF receptors (FGFRs) and heparan sulfate proteoglycan (HSPG) co-receptors. We show that the latter experimental modality is instructive for model selection using a Bayesian parameter inference. Our results provide novel insights into how different receptor tyrosine kinase (RTK) systems differentially wire the MAPK network to fine-tune fate decisions at the cell population level.**

**Keywords**  cell fate determination; ERK signaling dynamics; mechanistic modeling; microfluidics; parameter estimation

**Subject Category**  Signal Transduction

**Mol Syst Biol. (2019) 15: e8947**

## Introduction

Signaling dynamics, rather than steady states, have been shown to control cell fate responses (Levine *et al*, 2013). For multiple systems including receptor tyrosine kinase signaling (RTK), signaling heterogeneity can explain the fate variability observed within a cell population (Cohen-Saidon *et al*, 2009; Chen *et al*, 2012). Both biological noise extrinsic to individual cells and intrinsic variability within signaling networks shape the cell fate. It has been proposed that the dynamic nature of signal transduction enables accurate information transmission in the presence of noise (Wollman, 2018). Measuring single-cell signaling dynamics is therefore key to understanding how cellular responses correlate with specific cell fate decisions.

The extracellular signal-regulated kinase (ERK) is a key regulator of fates such as proliferation and differentiation. It functions within a mitogen-activated protein kinase (MAPK) signaling pathway in which growth factor (GF) receptors activate a membrane-resident Ras GTPase that subsequently triggers a MAPK cascade leading to ERK activation (Avraham & Yarden, 2011). Rat adrenal pheochromocytoma PC-12 cells have been widely used as a model system to study the regulation of cell fate by MAPK signaling (Marshall, 1995). Stimulation with EGF or NGF leads to population-averaged transient or sustained ERK states, which specifically trigger proliferation or differentiation. Thus, ERK signal duration has been proposed as a key determinant of cell fate (Marshall, 1995; Santos *et al*, 2007). These distinct ERK states result from GF-dependent control of the MAPK network (Santos *et al*, 2007), with negative and positive feedback producing all-or-none adaptive or bistable outputs, respectively (Xiong & Ferrell, 2003; Santos *et al*, 2007; Avraham & Yarden, 2011). More recently, single-cell assays have indicated that EGF/NGF induces heterogeneous dynamic signaling states across a cell population (Ryu *et al*, 2015). While EGF leads to transient ERK activity responses, NGF induces transient or sustained responses in an isogenic population due to variability in expression of signaling components and receptor-dependent modulation of the negative and positive feedback loops. This might explain how NGF can induce a heterogeneous mix of differentiating and proliferating cells (Chen *et al*, 2012). Further support that dynamic ERK signaling states control fate decisions stems from model-based prediction of dynamic GF stimulation schemes that induce synthetic ERK activity patterns that determine fate decision independently of GF identity (Ryu *et al*, 2015).

An additional GF, FGF2, also activates ERK through FGF receptors (FGFRs) and regulates processes such as angiogenesis, wound healing, and development (Ornitz & Itoh, 2015). Upon FGF2 stimulation, FGFR dimerizes, autophosphorylates, recruits adaptors, and activates the Ras/RAF/MEK/ERK cascade (Ornitz & Itoh, 2015). In

1   Institute of Cell Biology, University of Bern, Bern, Switzerland
2   Department of Biosystems Science and Engineering, ETH Zurich, Basel, Switzerland
3   Institute of Advanced Machinery and Design, Seoul National University, Seoul, Korea
    *Corresponding author. Tel: +41 31 631 46 37; E-mail: olivier.pertz@izb.unibe.ch
    †These authors contributed equally to this work
    ‡Present address: Research Laboratory of Electronics, Massachusetts Institute of Technology, Cambridge, MA, USA

PC-12 cells, FGF2 induces sustained ERK activity, which correlates with differentiation (Qui & Green, 1992). FGF–FGFR interactions are further regulated by a heparan sulfate proteoglycan co-receptor (HSPG) (Ornitz, 2000; Matsuo & Kimura-Yoshida, 2013). FGF2 initially binds to HSPGs through a high-affinity interaction, followed by a $2^{nd}$ lower affinity interaction leading to a HSPG/FGF2/FGFR trimeric complex. The latter subsequently dimerizes to a dimer of trimer complex that can autophosphorylate and signal downstream (Ornitz & Itoh, 2015). In marked contrast to signaling systems that exhibit sigmoidal dose responses, FGF2 elicits a biphasic dose response of signaling and cell fate outputs, where an intermediate concentration of FGF2 elicits higher activation of signaling and fate outputs compared to low and high FGF2 concentrations. For example, bell-shaped neuronal differentiation (Williams *et al*, 1994), or cell proliferation fate outputs (Zhu *et al*, 2010) are observed in FGF2 dose–response challenges in different cell systems. This correlates with a biphasic dose response of ERK activity outputs (Zhu *et al*, 2010; Kanodia *et al*, 2014). The ability of FGF2 to induce biphasic dose responses has been proposed to emerge from competition of FGF2 binding to HSPGs and the FGFR (Kanodia *et al*, 2014). However, the FGF2-dependent signaling network has been significantly less defined than the network downstream of EGF and NGF.

An important question in the signaling field is how different RTKs can specify different cell fates by using the MAPK network. Here, we explore how FGF2 controls ERK activity dynamics at the single-cell level in PC-12 cells. We find that FGF2 induces a mix of dynamic ERK states that are distinct from those of EGF/NGF. An increase in FGF2 input gradually modulates the distribution of transient/sustained ERK states. Using microfluidics to temporally perturb the MAPK signaling network, we further explore the logic behind these different signaling states. Our data together with mathematical modeling show that the FGF2-dependent MAPK signaling network underlying these responses consists of an extracellular FGF2/FGFR/HSPG interaction layer coupled to an intracellular MAPK network layer with a simple negative feedback. We conclude that EGF, NGF, and FGF2 wire the MAPK network differently to induce distinct population distributions of ERK states that fine-tune fate decisions at the cell population level. Our data therefore provide new insights into how different RTKs decode binding of their cognate GF by engaging distinct MAPK network structures. Our results suggest that the FGF2/MAPK signaling network has evolved to translate increasing FGF2 inputs into gradual changes in the population distribution of dynamic ERK states. This might be important to regulate fate decisions during the interpretation of morphogen gradient.

# Results

### FGF2 induces dynamic signaling states distinct from those induced by EGF/NGF

EGF/NGF-triggered ERK activity responses have been widely studied in PC-12 cells. However, single-cell studies have revealed a much higher signaling complexity than previously anticipated (Ryu *et al*, 2015). Here, we asked if FGF2 potentially induces ERK activity dynamics within a cell population that are distinct from those of EGF/NGF. To study FGF2 signaling at the single-cell level, we used

a PC-12 cell line stably expressing EKAR2G, a fluorescence resonance energy transfer (FRET)-based biosensor for endogenous, cytosolic ERK activity. EKAR2G has been extensively validated elsewhere (Harvey *et al*, 2008; Fritz *et al*, 2013). To extract single-cell temporal ERK activity patterns, we used a CellProfiler-based (Kamentsky *et al*, 2011) image analysis pipeline for segmentation and tracking of single cells, and for computation of a per-cell average FRET biosensor ratio. We used a computer-programmable microfluidic device to temporally perturb cells using GF pulses (Fig 1A).

First, we stimulated cells with a typical EGF/NGF/FGF2 concentration of 25 ng/ml (Fig 1B and C). We used a fluorescent dextran for quality control of the GF delivery by the microfluidic device (Fig 1C, lower red trace). As expected, when we evaluated population-averaged temporal ERK activity patterns, EGF led to transient ERK activity, while NGF induced a peak followed by sustained ERK activity with an amplitude lower than the peak. In contrast, FGF2 led to a transient ERK peak that was sharper than the one evoked by EGF. After this fast adaptation, ERK activity gradually increases over time. Since increasing FGF2 concentration induces a biphasic dose response in fate determination and ERK activity in a variety of cell systems (Zhu *et al*, 2010; Kanodia *et al*, 2014), we also tested such increase in our system. We stimulated PC-12 cells with EGF/NGF/FGF2 concentration in a 0.25–250 ng/ml range, as in previous works (Zhu *et al*, 2010; Kanodia *et al*, 2014; Fig 1D). On average, all EGF concentrations triggered an initial ERK peak with identical amplitude but with faster adaptation at higher GF concentrations. In contrast, 0.25 ng/ml NGF only induced moderate ERK activity without an initial ERK activity peak. 2.5 ng/ml NGF led to sustained ERK activity after a small initial peak. 25 and 250 ng/ml led to almost indistinguishable profiles of an ERK activity peak followed by sustained ERK activity. FGF2 stimulation led to different population-averaged temporal ERK activity patterns than both EGF and NGF. Indeed, 0.25 ng/ml FGF2 led to sustained ERK activity without a robust initial peak, whereas 2.5, 25, and 250 ng/ml FGF2 led to a clearly defined initial ERK transient. At 25 and 250 ng/ml FGF2, after the initial transient, we again observed slow ERK activity recovery. The previously described biphasic dose response of ERK activity is evident when we consider a time point after the initial ERK activity peak (Zhu *et al*, 2010; Kanodia *et al*, 2014). Additionally, we observe that the amplitude of the $1^{st}$ ERK peak activity was highly similar across GF identity/concentration (Fig EV1A).

### FGF2 dose response leads to an ERK activity population distribution that is wider than that associated with EGF and NGF

Heterogeneous single-cell dynamic signaling states were evident when single-cell temporal ERK activity patterns were overlaid over population-averaged temporal ERK activity patterns (Fig EV1B). To examine this heterogeneity, we pooled all trajectories (EGF, NGF, and FGF2—4 concentrations) using a time interval ranging from shortly before GF stimulation to 60′ after stimulation (Fig 1E). We then applied hierarchical clustering with dynamic time warping (DTW) (Giorgino, 2009) to extract classes of single-cell temporal ERK activity patterns. DTW calculates similarity between two time series by matching shape features that may be shifted in time between the two series. Visual inspection of the dendrogram obtained from this procedure led us to identify 6 major dynamic

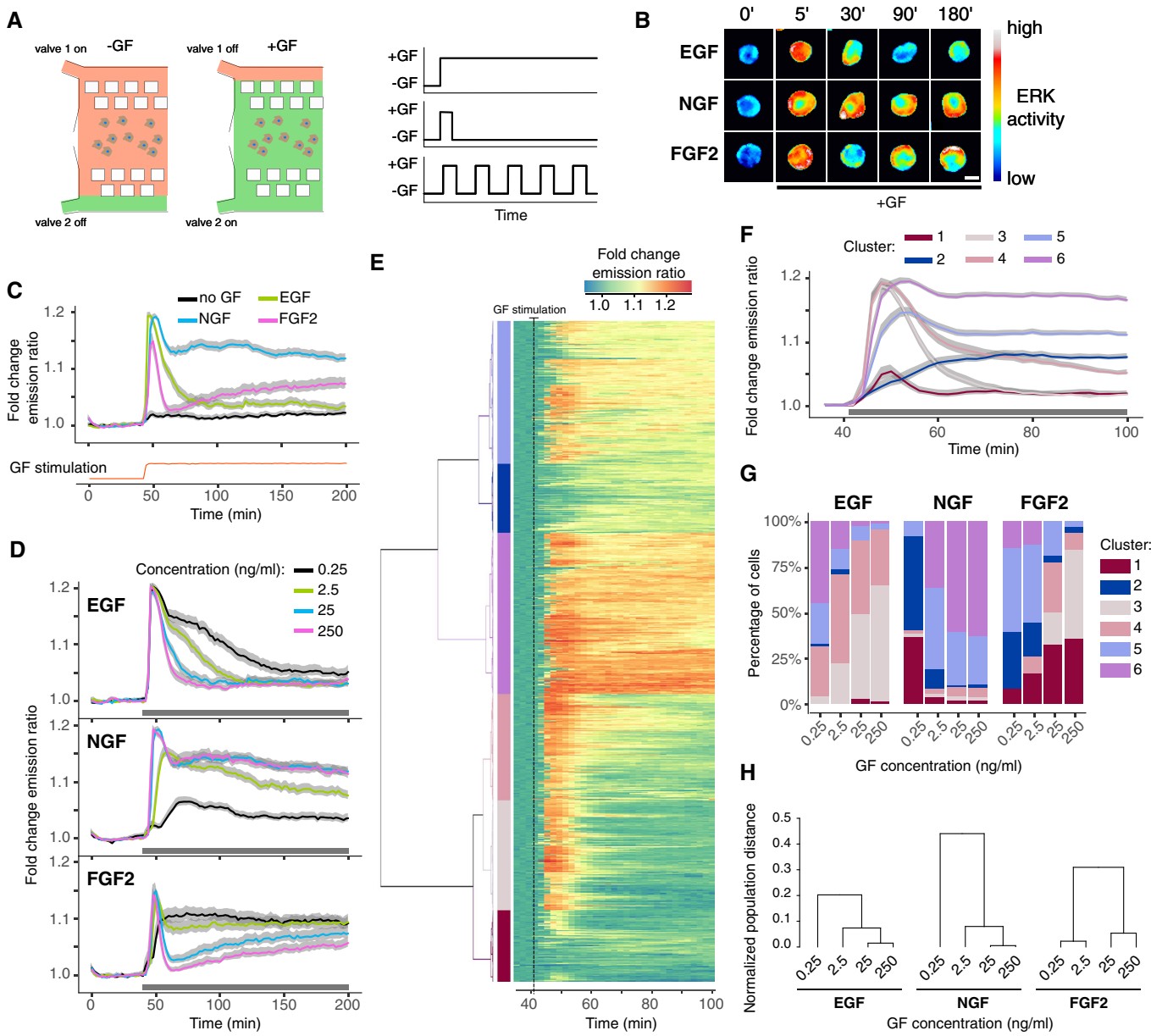

**Figure 1. FGF2 induces different dynamic ERK activity signaling states than EGF/NGF.**

A　Flow-based, microfluidic device for temporal GF delivery. Computer-controlled, pressure pump enables mixing of medium and GFs to deliver GF pulses in cells cultured in the microfluidic device. The right panel illustrates typical GF stimulation patterns.

B　Representative EKAR2G ratio images of cells treated with 25 ng/ml EGF, NGF, and FGF2. Ratio images are color-coded so that warm/cold colors represent high/low ERK activation levels. Scale bar = 50 μm.

C, D　Population averages of ERK activity dynamics in response to stimulation with 25 ng/ml (C), or with a dose–response challenge using 0.25, 2.5, 25, and 250 ng/ml EGF, NGF, and FGF2 (D). Single-cell time series were normalized to their own means before GF stimulation, $t = [0, 40]$. Red curve at the bottom of panel C indicates GF stimulation profile measured simultaneously using an Alexa 546-labeled dextran. $N = [48, 120]$ cells per GF concentration. ERK dynamics measured at 2′ intervals.

E　Hierarchical clustering of pooled ($N = 983$) single-cell time series from panel (D). To focus on relevant ERK dynamics, we trimmed x-axis to $t = [36, 100]$ min. Each row of the heatmap corresponds to a time series of a single cell. We used dynamic time warping and Ward's linkage method for building the dendrogram, which was then cut to distinguish 6 clusters that are color-coded on the left.

F　Average ERK activity across 6 clusters identified in panel (E), color-coded as in (E).

G　Distribution of ERK activity trajectories across 6 clusters from panels (E and F) in response to different GF dosages.

H　Separability between populations of single-cell trajectories calculated as normalized area under the curve of Jeffries–Matusita distance along time (Materials and Methods, Appendix Fig S2B). The dendrogram was created using the complete-linkage method.

Data information: In panels (C, D, and F), gray band indicates 95% CI for the mean, representative of 3 replicates. In panels (D and F), black horizontal bar indicates GF stimulation.

Source data are available online for this figure.

patterns/clusters, which are highlighted by vertical color bars in Fig 1E. Figure 1F summarizes population averages for each cluster, while Fig EV1C displays single-cell temporal ERK activity patterns for each cluster. Even though all clusters differ in amplitude, we can recognize adaptive behavior in clusters 1, 3, and 4, and sustained activation in clusters 2, 5, and 6. We then computed the population distribution of these representative single-cell temporal ERK activity patterns across all experimental conditions (Fig 1G). Low-amplitude adaptive and sustained ERK activities (clusters 1 and 2) were largely absent from responses to all EGF stimulations, indicating robust ERK signaling. With the increase of EGF, an adaptive cluster 3 replaced sustained clusters 5 and 6. In contrast, the lowest NGF dose induced a mix of low-amplitude adaptive and sustained responses. High NGF concentrations induced high-amplitude sustained clusters 5 and 6 with a decreasing contribution of intermediate responses. We observed a wider mix of cluster distribution for FGF2 dose response. 0.25 ng/ml FGF2 led to a mix of low- and high-amplitude sustained responses. 2.5 ng/ml FGF2 decreased sustained responses in favor of cluster 4 (high amplitude, intermediate adaptation). Then, with an increased FGF2 dosage the distribution shifted to strongly adaptive responses with low and high amplitudes.

Intrigued by the fact that FGF2 induces slow ERK activity recovery after the $1^{st}$ peak at 25 and 250 ng/ml in population-averaged measurements (Fig 1D), we tested if this was also the case at the single-cell level. For that purpose, we repeated the clustering analysis individually for EGF, NGF, and FGF2 dose–response experiments on a time ranging from shortly before GF addition to 160′ after stimulation (Fig EV1D). For FGF2, this again identified single-cell temporal ERK activity patterns that displayed a robust 1st ERK activity peak followed by different levels of adaptation (clusters 1–3), or sustained ERK activity. Importantly, the three adaptive clusters that were present at high FGF2 concentrations displayed slow ERK activity recovery after adaptation. This specific phenomenon is not present in EGF/NGF dose responses.

To independently assess that FGF2 evokes a distinct and wider mix of single-cell temporal ERK activity patterns than EGF/NGF in a dose response, we applied PCA decomposition and calculated accumulated pairwise distances of the response distribution at different time points (Figs 1H and EV2). Both approaches showed that single-cell temporal ERK activity pattern population distributions for different GF dosages are more separated for FGF2 (Appendix Text).

### Decoding FGF2/MAPK signaling network properties by temporal perturbation of ERK dynamics

We then sought to identify the signaling network structure that explains how the FGFR/MAPK network evokes ERK states different from those evoked by EGF/NGF. For that purpose, we dynamically perturbed cells by delivering single or multiple GF pulses of different lengths and concentrations using our microfluidic device (Fig 1A). This approach captures salient features of the MAPK network not accessible with sustained GF stimulation and, in many cases, induces population-homogeneous signaling states that are simpler to interpret (Ryu et al, 2015). We stimulated PC-12 cells with pulses of 3′, 10′, and 60′ with the four concentrations of each GF used previously. We plotted the population-averaged temporal ERK

activity patterns (Fig 2) and used hierarchical clustering to extract representative dynamic patterns for each GF pulse pattern (Fig EV3).

The pulsed EGF/NGF dose responses were consistent with our previous observations (Ryu et al, 2015). Population-averaged temporal ERK activity patterns exhibited a full-amplitude initial ERK activity peak followed by robust adaptation for all EGF concentrations for 3′ or 10′ pulse, except for a 3′ 0.25 ng/ml EGF pulse (Fig 2A) where the peak was less pronounced. The 60′ EGF pulse revealed distinct adaptation kinetics after the initial ERK activity peak with faster adaptation at higher EGF dose. As observed in sustained stimulation, full adaptation occurred concomitantly with EGF washout. Clustering of single-cell temporal ERK activity patterns revealed adaptive responses across the EGF doses and pulsing schemes (Fig EV3A). In the case of NGF, the 0.25 ng/ml concentration did not yield ERK activation across any pulsing scheme (Fig 3B). Above this concentration, high NGF input (achieved by increasing dose and/or pulse duration) gradually shifted the population-averaged temporal ERK activity patterns from transient to more sustained profile. Clustering of single-cell temporal ERK activity patterns revealed a mix of transient and sustained responses, with sustained clusters contributing more at high NGF inputs (Fig EV3B).

Intriguingly, varying FGF2 dosage and pulse duration again revealed more complex population-averaged temporal ERK activity patterns than for EGF/NGF (Fig 2C). At a threshold input, we observed a new dynamic pattern, whereby an initial adaptive ERK activity peak was followed by a rebound that then decayed slowly. This dynamic pattern was visible at 25 ng/ml 10′ pulse and at 250 ng/ml 3′ and 10′ pulse. Lower or shorter FGF2 dosages induced only transient ERK activities. Clustering revealed a transient cluster as well as the characteristic ERK activity with a rebound (Fig EV3C). The latter cluster was enriched at high FGF2 inputs.

The 60′ FGF2 pulse led to even more complex population-averaged temporal ERK activity patterns. The pattern with ERK activity rebound emerged at 2.5–250 ng/ml FGF2, and for these concentrations, the rebound ensued only after GF was washed away. The adaptation after the initial peak was stronger at higher GF dosages. In contrast, at the lower concentration of 0.25 ng/ml FGF2, we observed sustained activation during the GF pulse and a slow decay after GF washout. Clustering of responses to 60′ pulse confirms that the lowest FGF2 dosage induces high and low sustained responses without a rebound, while higher GF concentrations result in a much stronger adaptation after the initial peak.

To probe network architectural features that might work at longer timescales and to test how the MAPK network responds to novel GF inputs before full adaptation, we subjected PC-12 cells to multiple 3′ pulses separated by 20′ pauses (Fig 3A). Multiple pulses of EGF led to transient population-averaged temporal ERK activity patterns with 20′ timescale adaptation that were in phase with the pulse pattern (Fig 3A). As previously shown (Ryu et al, 2015), increasing EGF concentrations correlated with increased ERK activity peak amplitude desensitization over the timescale of hours. Clustering confirmed homogeneous single-cell temporal ERK activity patterns across the population (Fig 3B). Multi-pulse 3′-20′ NGF also led to transient population-averaged temporal ERK activity patterns that were in phase with the stimulation pattern (Fig 3C). Again, as previously described (Ryu et al, 2015), desensitization occurred at the timescale of hours, but this was not dependent on the NGF

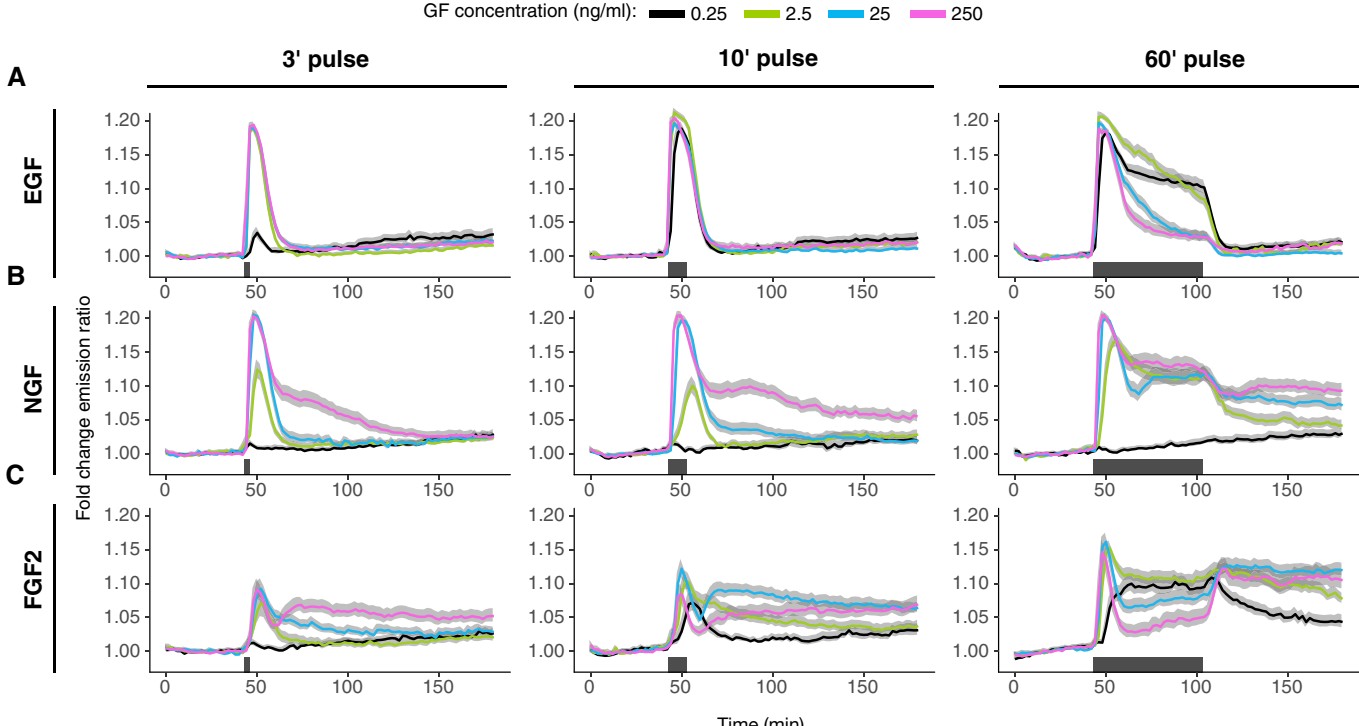

**Figure 2.** ERK activity dynamics in response to single-pulse stimulation.

A–C  Population average of ERK activity dynamics in response to 3′, 10′, and 60′ EGF (A), NGF (B), and FGF2 (C) single-pulse stimulation. Single-cell time series were normalized to their own means before the GF stimulation, $t$ = [0, 40]. Solid lines—population mean, $N$ = [39, 166], replicates: EGF: 1, NGF: 1, FGF: representative of 3 replicates; gray bands—95% CI for the mean; black horizontal bars—duration of GF stimulation.

Source data are available online for this figure.

concentration as for EGF. Additionally, adaptation of individual ERK activity pulses weakened with an increased NGF dosage. Clustering indicated that this phenomenon emerged from a mix of cells with different adaptive strengths, with weakly adaptive subpopulation dominating higher NGF concentrations (Fig 3D).

In contrast, FGF2 multi-pulse datasets revealed distinct population-averaged temporal ERK activity patterns (Fig 3E, Movie EV1) than those seen with EGF and NGF. Consistently, with the single-pulse data, 0.25 ng/ml FGF2 pulses did not activate ERK, while 2.5 ng/ml FGF2 pulses did induce adaptive ERK activity peaks in phase with the GF stimulation. 25 ng/ml FGF2 led to an initial peak followed by sustained ERK activity of amplitude lower than that of the initial peak. 250 ng/ml FGF2 pulses led to ERK activity peak immediately followed by short adaptation and a rebound phase, ultimately leading to sustained ERK activity. Re-triggering with the second pulse then led to immediate adaptation, followed by recovery to sustained ERK activity levels. Thus, from the 2nd pulse on, ERK activity was anti-phasic with respect to the stimulation pattern. To evaluate single-cell temporal ERK activity patterns associated with population averages, we used hierarchical clustering with Euclidean distance to preserve information about any potential phase shift (Fig 3F). We also omitted the 0.25 ng/ml FGF2 dataset to avoid non-responding cells. We identified four clusters, two of which resembled the in-phase ERK activity induced by the 2.5 ng/ml FGF2 (clusters 1 and 2), while the remaining two resembled the

anti-phase ERK activity evoked by the 250 ng/ml FGF2 (clusters 3 and 4). Indeed, 2.5 ng/ml FGF2 consists of in-phase clusters, while 250 ng/ml FGF2 is dominated by anti-phase single-cell temporal ERK activity patterns. 25 ng/ml FGF evoked a 35–65% population distribution of in- and anti-phase single-cell temporal ERK activity patterns, explaining the emergence of sustained ERK activity in population-averaged measurements. Together, these datasets indicate the existence of different MAPK network circuitries downstream of the three GFs, with FGF2 being able to evoke more distinct signaling states than EGF/NGF (e.g., a phase shift at high FGF2 concentrations). The FGF2 concentration-dependent gradual emergence of the ERK phase shift in multi-pulse experiments parallels FGF2's ability to gradually shift the population distribution of transient/sustained ERK states in response to sustained stimulation.

## Evaluating the role of HSPGs in the FGF2 signaling responses

Heparan sulfate proteoglycans are important modulators of FGF2 signaling and enable a biphasic dose response of signaling outputs (Kanodia *et al*, 2014). To evaluate the role of HSPGs, we took advantage of the widely used chlorate ($NaClO_3$) treatment to inhibit HSPG sulfation, and thus binding of FGF2 to HSPGs (Ornitz & Itoh, 2015). We benchmarked the effect of that perturbation against the FGF2-specific gradual phase shift in ERK activity in response to increasing input in a multi-pulse experiment. We gradually inhibited

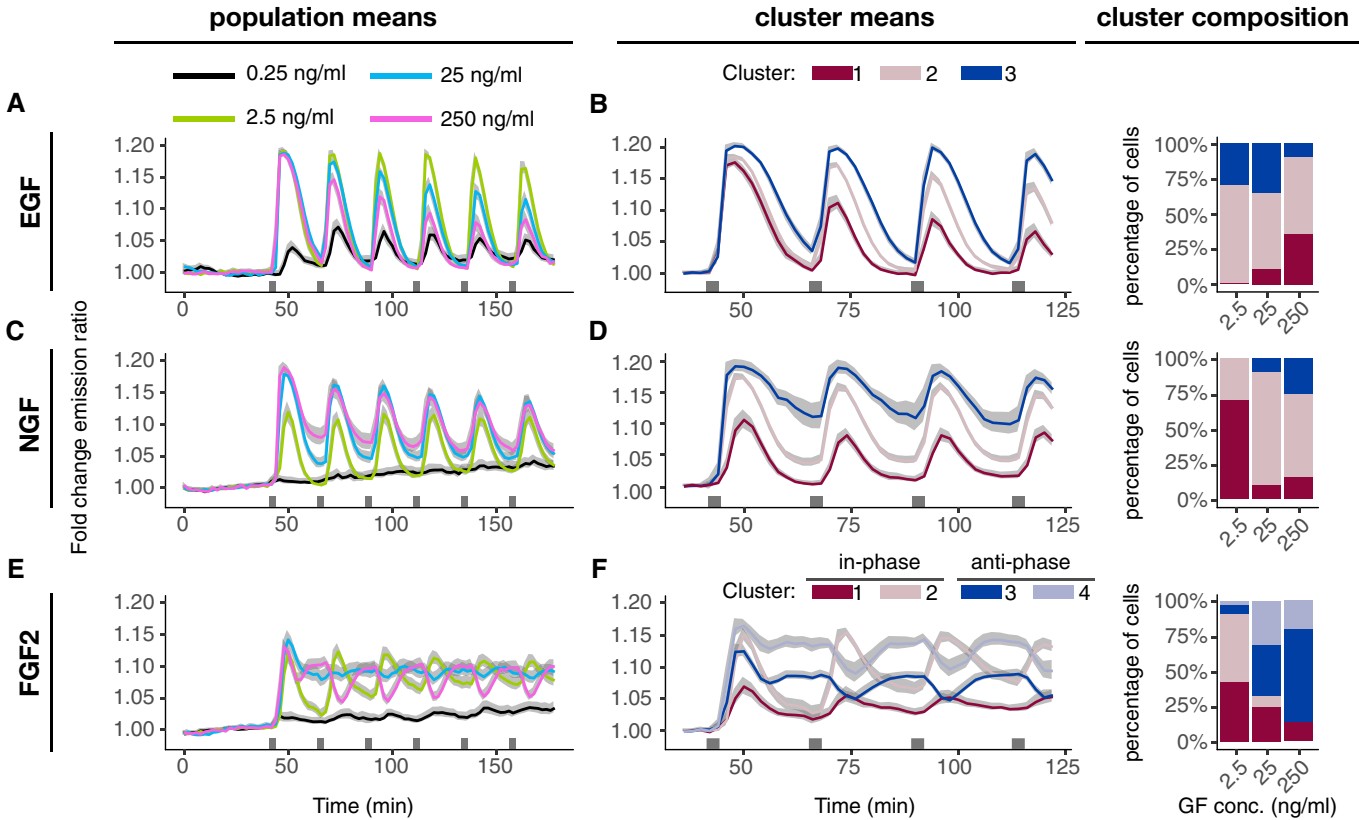

**Figure 3. ERK activity dynamics in response to multi-pulse stimulation.**

A  Population average of ERK activity dynamics in response to a multi-pulse 3′–20′ EGF stimulation. Single-cell time series were normalized to their own means before the GF stimulation, $t$ = [0, 40].
B  Cluster averages of ERK activity and distribution of single-cell trajectories across clusters.
C  Population average of ERK activity dynamics in response to a multi-pulse 3′–20′ NGF stimulation. Single-cell time series were normalized to their own means before the GF stimulation, $t$ = [0, 40].
D  Cluster averages of ERK activity and distribution of single-cell trajectories across clusters.
E  Population average of ERK activity dynamics in response to a multi-pulse 3′–20′ FGF2 stimulation. Single-cell time series were normalized to their own means before the GF stimulation, $t$ = [0, 40].
F  Cluster averages of ERK activity and distribution of single-cell trajectories across clusters.

Data information: We performed hierarchical clustering with the Manhattan distance and the complete-linkage method; we cut the dendrogram at 3 (B, D) and 4 clusters (F) for visualization. Solid lines—population mean, $N$ = [52, 91], replicates: EGF: 1, NGF: 1, FGF: representative of 3 replicates; gray bands—95% CI for the mean; black horizontal bars—duration of GF stimulation.

Source data are available online for this figure.

HSPG sulfation with 10, 25, and 50 mM NaClO₃ and evaluated binding of a fluorescently labeled FGF2 to (un-)perturbed cells (Fig 4A and B). We observed increasing quanta of FGF2 fluorescence retained after each pulse in unperturbed cells, likely reflecting incremental HSPG binding, as well as endocytosed material. In contrast, gradual HSPG inhibition resulted in progressive loss of FGF2 binding to the cell, with almost no remaining FGF2 at 50 mM NaClO₃. Non-treated cells exposed to 2.5 ng/ml FGF2 pulses displayed the typical in-phase pattern. Gradual NaClO₃-mediated HSPG inhibition led to blunting of the ERK activity amplitudes that however remained in-phase, with only low residual ERK activity at 50 mM NaClO₃ (Fig 4C). While untreated cells exposed to 250 ng/ml FGF2 displayed the typical anti-phasic ERK activity, gradual NaClO₃-mediated HSPG inhibition progressively rescued in-phase ERK activity, at the same time decreasing ERK activity amplitude (Fig 4D). These results show that FGF2's ability to evoke ERK phase shifting

in multi-pulse dose–response experiments depends on HSPG/FGF2 interactions.

## Cell fate decision in response to sustained and pulsed GF stimulation

We then set out to correlate ERK activity dynamics with fate decisions. We repeated select sustained and pulsed dose–response experiments and evaluated the differentiation fate by quantifying neurite outgrowth using an automated image segmentation pipeline (Fig 5A and B). 0.25 ng/ml EGF led to low but still measurable levels of differentiation, which is consistent with its ability to induce slowly adapting, almost sustained ERK signaling (Figs 1D–G and EV1A). Higher EGF dosages that cause faster ERK adaptation resulted in lower cell differentiation. We observed low levels of differentiation at 0.25 ng/ml NGF, which is consistent with low

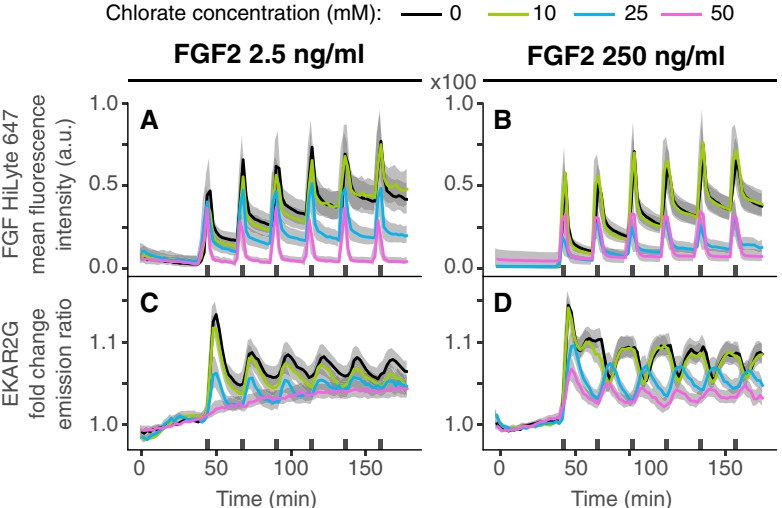

**Figure 4. ERK activity dynamics in response to HSPG perturbation.**

A–D  Whole-cell mean fluorescence intensity of labeled FGF2 (HiLyte 647) used in a dose–response challenge with 10, 25, and 50 mM of NaClO₃ and a multi-pulse 3′–20′ stimulation with FGF2 2.5 ng/ml (A, C), and 250 ng/ml (B, D). (C, D) Population average of ERK activity dynamics in response to the dose–response challenge. Solid lines—population mean, *N* =[43, 65] representative of 2 replicates; gray bands—95% CI for the mean; black bars—duration of GF stimulation.

Source data are available online for this figure.

amplitudes of single-cell temporal ERK activity patterns at this GF concentration. 2.5–250 ng/ml NGF led to potent differentiation, which is in line with largely sustained single-cell temporal ERK activity patterns. FGF2 induced a biphasic dose response in differentiation levels. At 0.25 ng/ml FGF2, we observed low levels of differentiation, which does not correlate with a mix of high- and low-amplitude sustained single-cell temporal ERK activity patterns (Fig 1D, F and G). 2.5 and 25 ng/ml FGF2, which consist of a mix of robust transient and sustained single-cell temporal ERK activity patterns, led to relatively high differentiation, however not to the level evoked by NGF. Finally, 250 ng/ml FGF2 dominated by transient single-cell temporal ERK activity patterns displayed lower levels of differentiation than 2.5 and 25 ng/ml FGF2. Except for 0.25 ng/ml FGF2, a qualitative correlation exists between single-cell temporal ERK activity patterns and the differentiation fate.

Similar to what we have shown for EGF/NGF (Ryu *et al*, 2015), we asked if induction of different signaling states through dynamic GF stimulation would allow us to manipulate cell fate decisions (Fig 5C and D). We reasoned that a 3′ pulse of 250 ng/ml FGF2 that leads to an ERK activity peak followed by a sustained rebound that lasts hours (Fig 2C) would result in high differentiation. In contrast, a 3′ pulse stimulation at a lower FGF2 dosage, which does not induce ERK activity rebound, should not induce differentiation. We indeed observed a correlation between this specific signaling state and differentiation. As expected, control 3′ pulsed EGF stimulation did not lead to differentiation, except for some low levels of differentiation at 250 ng/ml. We attribute this spurious differentiation to low levels of the remaining EGF that cannot be completely washed out after the pulse application by pipetting, thus recapitulating the 0.25 ng/ml EGF sustained stimulation results (Fig 5A and B). The 3′ 250 ng/ml NGF pulse also led to potent differentiation, which correlated with its ability to induce sustained ERK activity. These results indicate that sustained ERK signaling states, evoked by specific

sustained or pulsed GF stimulation, correlate with the differentiation fate for all three GFs.

**Modeling the FGF2-evoked ERK activity responses**

We then sought to find a minimal signaling network topology that could recapitulate the highly specific ERK signaling responses evoked by FGF2 stimulation. We reasoned that the salient features visible in dynamic ERK signaling states evoked by sustained and pulsed GF stimulation would discriminate among candidate model topologies. We used a Bayesian nested sampling (NS) inference method (Skilling, 2006) to compute the posterior distribution of parameter sets for a candidate model based on experimental training datasets, enabling us to infer parameter ranges that can best reproduce the data.

We postulated a set of minimal models that aim to explain the FGF2-induced ERK activity responses. The models differed with respect to receptor interactions and intracellular signaling topologies. Three models of FGF2/HSPG/FGFR interactions (Fig 6A) were as follows. (i) Simple activation model, whereby sequential FGF2–HSPG and FGF2–HSPG–FGFR interactions ultimately lead to formation of a FGF2–HSPG–FGFR dimer and subsequent FGFR activation. The increase in FGF2 input activates FGFR until all receptors saturate (Fig 6A, right panel). (ii) Competitive activation model, whereby FGF2 can also directly bind to FGFR, although this complex does not activate the receptor. Assuming that FGF2–FGFR binding occurs faster than the FGF2–HSPG binding, a biphasic dose response can emerge as previously proposed (Kanodia *et al*, 2014; Fig 6A, right panel). Low/medium FGF2 concentrations lead to FGFR activation, while high FGF2 dosage titrates FGFR, precluding the formation of signaling-competent FGF2–HSPG–FGFR complexes. (iii) Competitive joint-activation model. In contrast to the competitive activation model, the FGF2–FGFR complex is signaling-competent as

## sustained stimulation

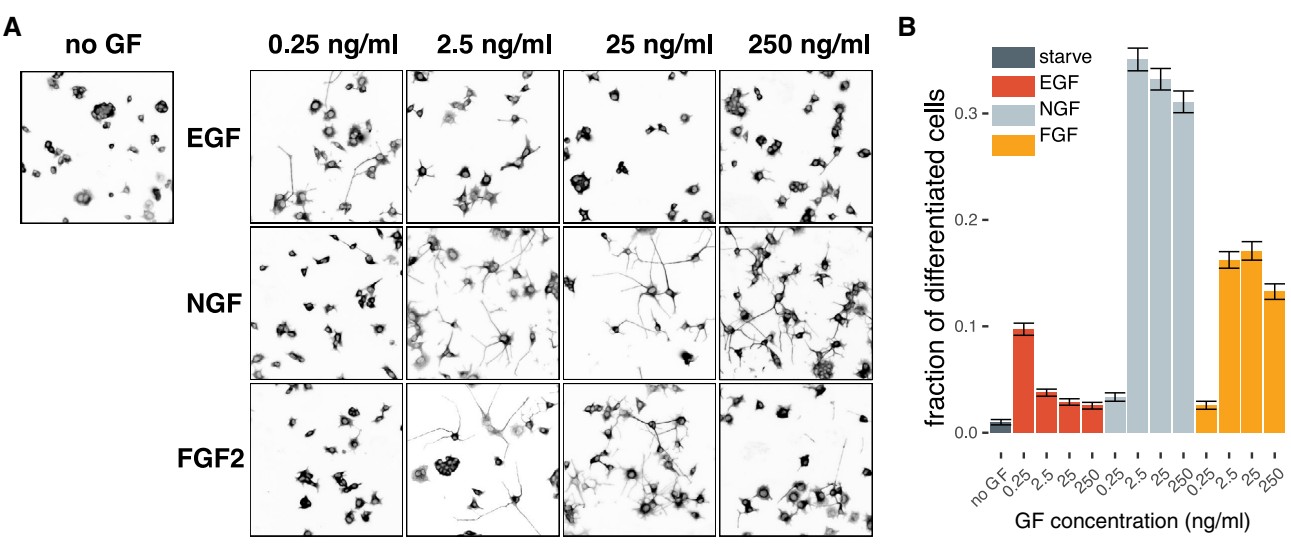

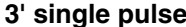

## 3' single pulse

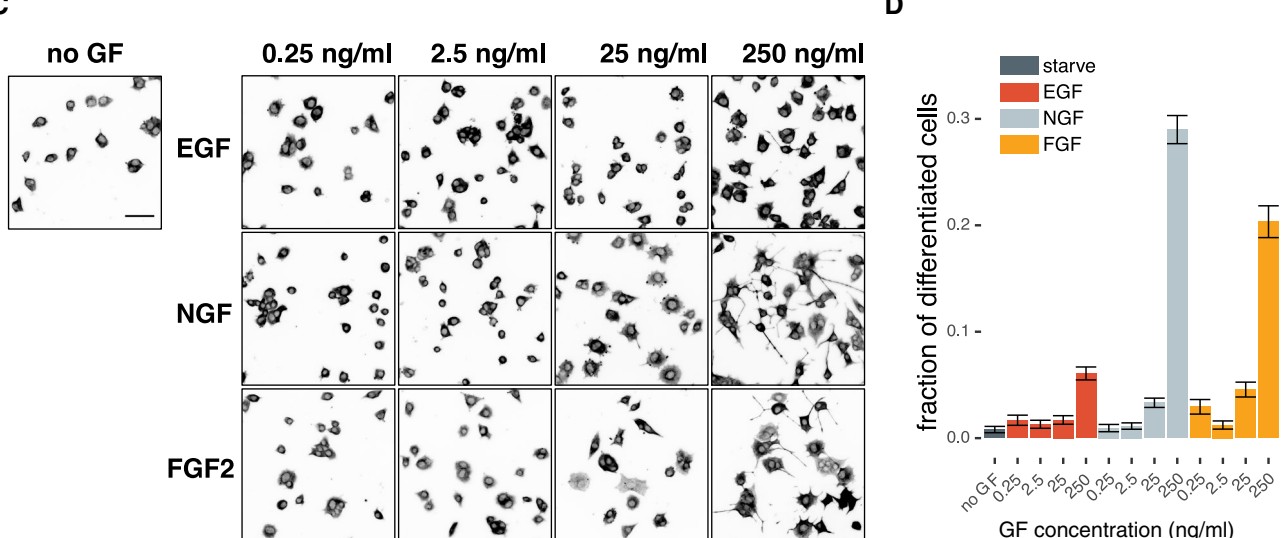

**Figure 5. Differentiation fate analysis in response to sustained/pulsed, GF dose–response stimulation.**

A   Representative images of differentiation experiments of sustained stimulation with respective GF. Cells are stained for alpha-tubulin and are shown in inverted black/white contrast.

B   Fraction of differentiated cells from sustained stimulation, calculated as fraction of cells with the total neurite outgrowth longer than the diameter of the cell soma using a CellProfiler-based automated image analysis routine. On average, 4,000 cells per condition were included in the analysis (min = 1,600, max = 12,500). Error bars indicate 95% CI for the mean.

C   Representative images of differentiation experiments of single 3-min pulse stimulation with respective GF. Cells are stained for alpha-tubulin and are shown in inverted black/white contrast. Scale bar = 50 μm.

D   Fraction of differentiated cells from 3-min single-pulse stimulation, calculated as fraction of cells with the total neurite outgrowth longer than the diameter of the cell soma using a CellProfiler-based automated image analysis routine. On average, 4,000 cells per condition were included in the analysis (min = 1,600, max = 12,500). Error bars indicate 95% CI for the mean.

Source data are available online for this figure.

previously proposed (Ornitz & Itoh, 2015). The signaling strength of FGF2–FGFR complex can be different from the FGF2–HSPG–FGFR complex, which allows residual levels of receptor activation, even at high FGF2 concentrations (Fig 6A, right panel).

For the intracellular signaling layer, we simplified the MAPK network by taking into account the Ras GTPase, as well as the Raf/MEK/ERK kinase cascade that itself allows the production of switch-like ERK activity (Fig 6B). We considered (i) a basic model without

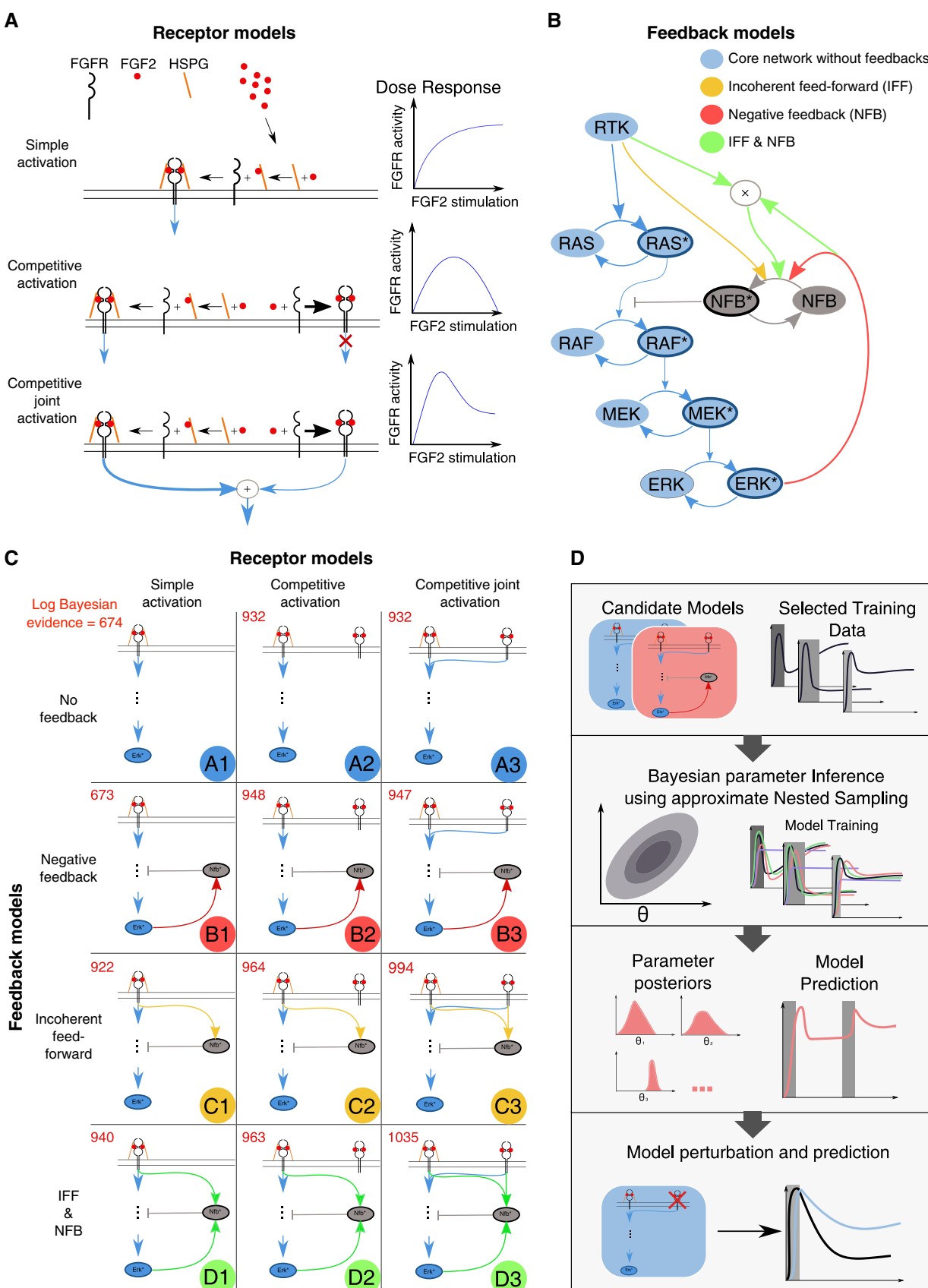

**Figure 6.**

**Figure 6. Description of candidate FGFR/FGF2 network models.**

A   Illustration of the three proposed receptor activation mechanisms. Simple activation: Only the FGFR–FGF2–HSPG complex activates downstream MAPK signaling (blue arrow). Competitive activation: the same as simple activation, but FGFR can bind FGF2 without HSPG. The resulting FGFR–FGF2 complex does not activate downstream pathways. Competitive joint activation: the same as competitive activation, but both FGFR–FGF2–HSPG and FGFR–FGF2 activate downstream MAPK signaling. Right panel indicates how FGFR activity will evolve upon FGF2 stimulation.

B   Proposed MAPK network feedback models. The minimal set of MAPK signaling nodes is shown, including a negative feedback regulator protein (NFB). Asterisks denote activated forms of these nodes. Basic system: no feedback or feed-forward structure. Incoherent feed-forward: Activation of the negative regulator is proportional to FGFR activation. Negative feedback: Activation of the negative regulator is proportional to ERK activation. Incoherent feed-forward and negative feedback: Activation of the negative regulator is proportional to the product of RTK activation and ERK activation.

C   A systematic overview of all proposed model topologies. The models are labeled A–D for different intracellular feedback structures, and 1–3 for the different receptor models.

D   An overview of our model selection procedure. We start with our set of 12 candidate models and a training set of experimental observations. For each of the candidate models, we perform Bayesian parameter inference using the approximate nested sampling algorithm. We obtain a posterior distribution of the parameters for each candidate model. We then subsequently benchmark models/associated parameter spaces for their ability to predict ERK activity dynamics for unobserved pulsed stimulation schemes. Finally, we further evaluate the ability of the candidate models to predict a biological perturbation (HSPG perturbation).

any feedbacks. We then added three different wirings to get the following models: (ii) a generic negative feedback from ERK to Raf as previously proposed (Santos *et al*, 2007); (iii) an incoherent feed-forward (IFF) system in which RTK modulates an intermediate negative regulator of Raf, termed negative feedback protein (NFB), based on our recent finding (Ryu *et al*, 2015); and (iv) a hybrid system in which the negative regulator of Raf is modulated by both an RTK-based IFF and a negative feedback from ERK to Raf in a multiplicative way (Ryu *et al*, 2015). While there is some support in the literature for the existence of positive feedback within the ERK signaling cascade (Santos *et al*, 2007), we first explored previously described negative feedback topologies that were proposed for the FGF2/MAPK signaling network (Kanodia *et al*, 2014), and investigated their interplay with the different receptor models.

By combining the three extracellular layer receptor models with the four intracellular MAPK network feedback models, we obtained a total of 12 candidate models (Fig 6C). The models consist of 9–15 species and up to 39 parameters. To account for the nonlinear nature of FRET measurements, we explicitly modeled the nonlinear relationship between the ratiometric output of the biosensor and the underlying activity of ERK as previously described (Birtwistle *et al*, 2011). The iterative steps of our modeling approach are depicted in Fig 6D and described below.

To avoid overfitting, we restrained the training datasets to four experimental conditions that capture dynamic features specifically relevant to FGF2/MAPK signaling: (i) Sustained 2.5 ng/ml FGF2 exhibits only low level of post-ERK activity peak adaptation (Figs 1D and EV1C); (ii) sustained 250 ng/ml FGF2 exhibits high level of post-ERK activity peak adaptation followed by slow recovery of ERK activity (Figs 1D and EV1C); (iii) 2.5 ng/ml FGF2 multi-pulse exhibits population-homogeneous ERK activity pulses in phase with the stimulus (Fig 3E); and (iv) 250 ng/ml FGF2 multi-pulse exhibits population-homogeneous anti-phasic ERK activity pulses (Fig 3E), and thus also captures the capability of ERK activity to rebound immediately after adaptation when a high concentration FGF2 pulse is applied. One rationale for focusing on these specific datasets was that our NS approach infers parameters on population-averaged ERK activity responses. We therefore avoided datasets that display highly heterogeneous single-cell ERK activity responses. These include stimulation with 0.25 ng/ml FGF2, which exhibits a mix of high- and low-amplitude single-cell ERK activity responses; or 25 ng/ml FGF2 multi-pulse stimulation, which results in a mix of in- and anti-phase single-cell ERK activity responses (Fig 3E and F).

For each of the 12 candidate models, we used NS to infer the parameters that explain the dynamic ERK activity observed in four training sets. We plotted the simulations for 500 sampled parameter vectors from the posterior as well as for the best parameter set (maximum likelihood estimate) and compared it to the corresponding experimental data (Fig EV4). Quantification of models' performance given experimental data with the Bayesian evidence (BE; Fig 6C, red number on top left of each model) excluded models A1 and B1 (which exhibited roughly a 280 orders of magnitude lower BE than the other models). Visual inspection of the 10 remaining models revealed that they faithfully reproduced many of the features of the training sets. The only model able to stringently explain all observations of the training dataset was the most complex model D3. The remaining models succeeded in reproducing the anti-phasic dose response for the pulse stimulation. Models A2, A3, C2, C3, and D2 failed to reproduce the initial peak, but succeeded in reproducing the ERK rebound upon sustained stimulation.

To further discriminate between the models, we benchmarked them against FGF2 stimulation patterns that were not used for training. With our microfluidic setup, we induced a 5′ single-pulse stimulation (Appendix Fig S1A), as well as a more complex multi-pulse stimulation (3′ pulse/30′ pause/20′ pulse/60′ pause/5′ pulse), at both 2.5 and 250 ng/ml FGF2 (Appendix Fig S1B). The latter induces ERK responses constrained by multiple feedbacks of the FGF2/MAPK signaling within one experiment, thus resulting in dynamics not seen in our previous experiments. For each of the 10 remaining models, we sampled 500 parameter sets from each posterior and plotted the simulated ERK activity using the best fit to the experimental dataset (Fig EV5). Since BE can only be computed for the dataset used for inference of the parameter posterior, we relied on visual inspection to evaluate the performance of the model fitting to the prediction dataset. This led us to exclude models that did not reproduce the general trends of the experimental datasets. This retained 4 models: B2, B3, C3, and D3, which are characterized by FGF2′s ability to directly bind FGFR in addition to canonical FGFR/HSPG interactions, as well as different intracellular feedback structures.

Given the importance of FGF2/HSPG interactions, we used the HSPG perturbation dataset to further discriminate the models. We set the HSPG concentration to zero and simulated the 4 remaining models using the maximum likelihood estimate of the training data (Fig 7A). The model B3 was the only one that recapitulated our experimental $NaClO_3$ perturbation (Fig 7B). Thus, this model

reproduced several features of the training data (Fig EV4), faithfully predicted unobserved ERK activity resulting from alternate dynamic stimulation (Fig EV5), and was the only one that predicted the effects of the HSPG perturbation (Fig 7A). Indeed, our approach remains coarse-grained to some extent since the slow ERK recovery after the sustained stimulation with 250 ng/ml FGF2 is captured with slightly different timescales (Fig EV4), suggesting that some additional mechanisms are at play.

While exploring MAPK models that are based on negative feedback yielded a satisfactory solution for explaining our data, the presence of sustained ERK activity in response to sustained stimulation with 2.5 ng/ml FGF2 suggested the possibility of a signaling network with positive feedback. To explore this, we formulated a range of new models that include positive feedback. The models E1, E2, and E3 included the three different extracellular receptor interaction schemes and a simple intracellular topology with positive feedback only (Appendix Fig S2A). The models B1′, C1′, and D1′ are variations of models B1, C1, and D1 shown in Fig 6C that include a positive feedback in addition to feedbacks explored previously (Appendix Fig S2B). In these models, we only focused on a simple activation modality in the receptor layer (Fig 6A). Our aim was to test whether the combination of negative/positive feedbacks would function in a similar context as EGF/NGF, of which the cognate receptors are not interacting competitively. Only some of the features of the training dataset could be recapitulated by models that included the positive feedback. Also, none of these models could produce a good fit to validation datasets (Appendix Fig S2). We therefore conclude that the positive feedback cannot explain FGF2′s ability to induce the observed ERK states.

# Discussion

An important question in the signaling field is how different RTKs, despite converging on a small number of core signaling processes (including MAPKs), can induce different fates (Lemmon & Schlessinger, 2010). An emerging concept is that the MAPK network is wired differently downstream of specific RTKs to generate distinct dynamic ERK states (Marshall, 1995; Santos *et al*, 2007). The latter are subsequently integrated into fate decisions through additional signal integration layers (Murphy *et al*, 2002; Gillies *et al*, 2017; Uhlitz *et al*, 2017). Here, we extend this notion by showing that FGF2, which has been poorly studied with respect to MAPK signaling dynamics, wires the MAPK network with a different logic than EGF/NGF, to induce distinct ERK states.

## FGF2 dose response gradually shifts the population distribution of ERK states

We report that FGF2 evokes strikingly different single-cell ERK states than EGF/NGF. Increasing FGF2 doses over 4 orders of magnitude gradually shifts the population distribution of ERK states from sustained toward adaptive response, albeit with a slow, long-term rebound (Fig 1E–G). This is consistent with the previously documented population-averaged biphasic dose responses of ERK activity and phenotypic outputs (Zhu *et al*, 2010; Kanodia *et al*, 2014). In contrast, increasing EGF doses trigger purely adaptive responses, and NGF produces both adaptive and sustained

responses, with a steep shift toward the latter at high input. These GF identity/concentration-dependent population distributions of ERK states are quantified using cluster decomposition (Fig 1G), PCA (Fig EV2A) and a distance metric to compare the separability of time courses (Figs 1H and EV2B and C). These results indicate that a single time-lapse recording of ERK activity from the sustained FGF2 dose–response challenge distinguishes low and high concentrations with a higher certainty than for EGF or NGF.

Pulsed FGF2 dose–response experiments also indicated FGF2′s ability to gradually induce distinct and wider ERK state population distributions than EGF/NGF. Increasing levels of pulsed FGF2 gradually led to robust switch-like activation and strong adaptation, followed by a clear rebound leading to sustained ERK activity (Figs 2C and EV3C), with adaptation remaining for the pulse duration in the 60′ pulse experiment (Figs 2C and EV3C). In the multi-pulse dose–response experiment, increasing FGF2 gradually led to a phase shift of ERK activity patterns relative to the GF pulse (Fig 3) that depended on HSPG interactions (Fig 4), while EGF/NGF was unable to produce such a phase shift. These results again highlight FGF2′s ability to gradually shift the ERK state population distribution and thus to translate increasing GF inputs into more clearly distinct signaling states than EGF/NGF.

## Mechanistic insight into the FGF2/MAPK signaling network

To understand the network circuitry underlying FGF2-evoked ERK signaling states, we took advantage of the salient features evident in our array of sustained/pulsed experiments combined with a mathematical modeling approach. Our model selection approach consists of the following steps (Fig 6D): formulation of minimal models that capture the relevant biology of the signaling system using *a priori* knowledge; carrying out Bayesian NS inference of the parameter space for each candidate model upon training on information-rich ERK states using temporal perturbations; and benchmarking model performance by predicting unknown stimulation schemes not used for training, and HSPG perturbation. We identified a simple network topology that recapitulates the ERK states observed in all these experiments. The model consists of a competitive joint activation at the receptor level (both FGF2/HSPG/FGFR and FGF2/FGFR complexes contribute to signaling), as well as a negative feedback loop from ERK to RAF (Fig 7B)—a structure recurrent in many MAPK networks (Santos *et al*, 2007; Birtwistle & Kolch, 2011). Thus, an interplay between a ligand, a co-receptor, and a receptor that leads to competitive activation of two signaling-competent complexes, coupled to a simple intracellular negative feedback, can recapitulate all observed FGF2-dependent ERK states.

To provide intuition about the interplay of the different molecular species involved in the network, we plotted their latent states. For the sake of simplicity, we focus on the robust, population-homogeneous, in- and anti-phase ERK states evoked by multi-pulse stimulation by 2.5 and 250 ng/ml FGF2 (Fig 7C). Schemes of the receptor interactions/activities are also provided for clarity (Fig 7D and E). At 2.5 ng/ml FGF2, FGF2 binds to FGFR and HSPG, increasing the HSPG+FGF2 and FGFR+HSPG+FGF2 concentration and leading to strong FGFR activity, as well as increasing the concentration of FGFR+FGF2 which leads to low FGFR activity (Fig 7D, top left panel). Upon FGF2 washout, FGF2 unbinds from the 3 receptor complex species. Thus, the FGF2 pulse initially leads

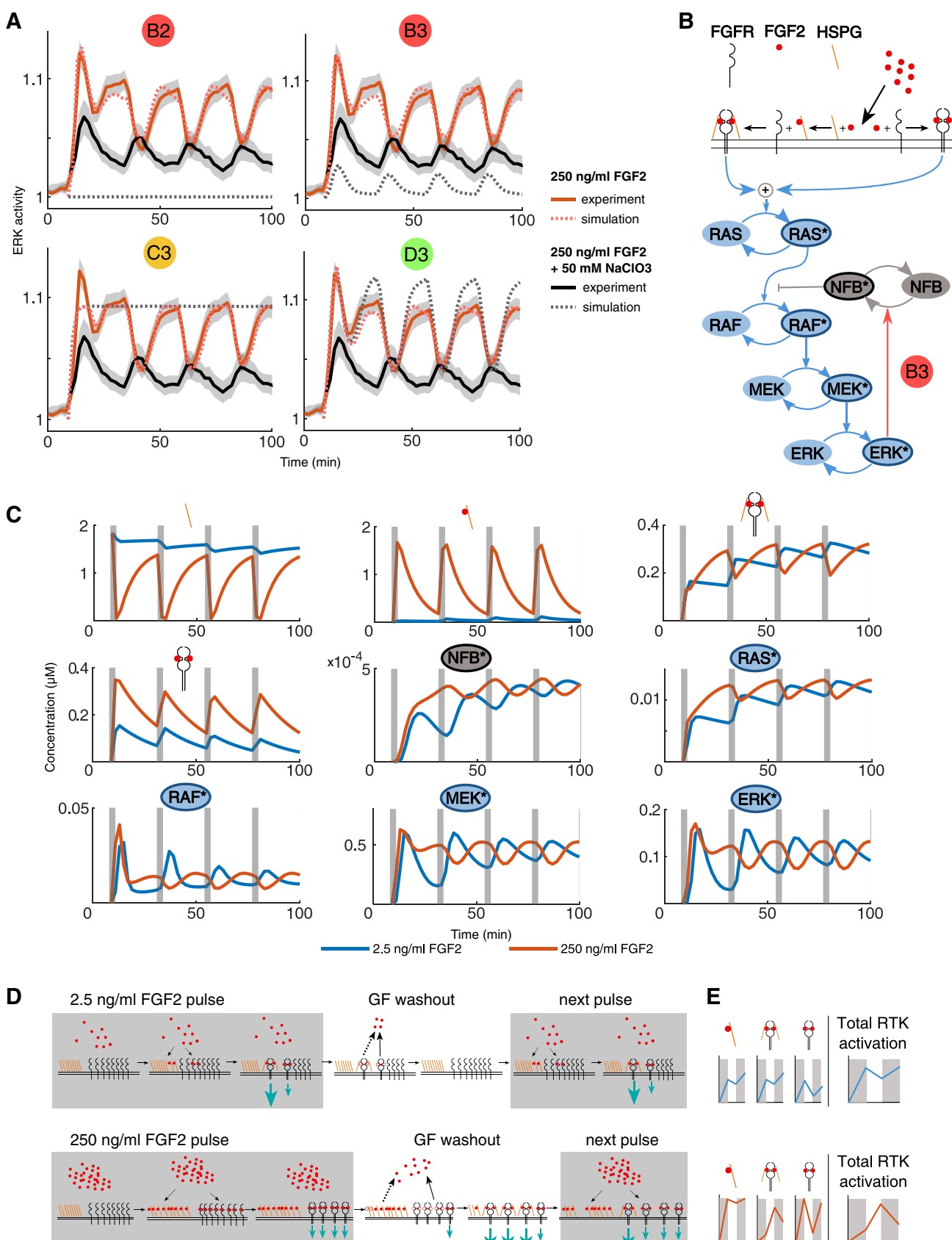

**Figure 7.**

**Figure 7.  Benchmarking models against HSPG perturbation, and description of model latent states.**

to combined high FGFR activity that subsequently adapts to some extent upon FGF2 washout, translating into a RAF/MEK/ERK activation/deactivation cycle with robust adaptation due to negative feedback. During subsequent FGF2 pulses, further increase in global FGFR activity is observed, explaining the in-phase ERK activation pattern. At 250 ng/ml FGF2, much more HSPGs will be FGF2-bound, mostly leading to the formation of HSPG+FGF2 and FGFR+FGF2 species at the cost of FGFR+HSPG+FGF2 complex formation. During the 1st pulse, global FGFR activation thus results from formation of some FGFR+HSPG+FGF2, but to a higher extent of FGFR+FGF2 complexes. Accordingly, RAF, MEK, and ERK get activated very similarly to the 2.5 ng/ml FGF2 pulse due to the switch-like activation enabled by the tripartite structure of the MAPK network. However, after this first pulse, FGF2 unbinds faster from FGFR than from HSPG (Fig 7D, bottom middle panel, plain and dotted arrows). The still abundant HSPG+FGF2 species are free to engage into FGFR+HSPG+FGF2 complexes, which induce strong FGFR activity. After initial adaptation due to strong negative feedback, FGFR activity leads to prolonged ERK activation. At the 2nd pulse, high FGF2 levels saturate HSPGs. Because FGF2 binding to FGFR is much quicker than HSPG-FGF2 to FGFR, a decrease of FGFR+HSPG+FGF2 complexes in favor of FGFR+FGF2 complexes ensues. This leads to an overall decrease in FGFR activity, followed by ERK inactivation (Fig 7C–E). An important feature captured by the model, which is already mentioned above, is that the HSPG+FGFR+FGF2 complex signals much more strongly than just FGFR+FGF2. This can be observed because Ras activity mostly follows the profile of HSPG+FGFR+FGF2, rather than that of FGFR+FGF2 species formation. Indeed, the model prediction that the HSPG+FGFR+FGF2 complex signals more strongly than the FGFR+FGF2 complex is expected since the latter has been proposed to be the canonical, stable signaling unit (Ornitz, 2000). These results indicate how the ERK activity phase shift emerges with increasing FGF2 concentrations through modulations of the abundance of different receptor species with differential signaling ability. We propose that the same receptor competition mechanism enables the gradual shift of the population distribution of transient/sustained ERK activity states in the sustained stimulation dose response.

## Distinct characteristics of specific RTK–MAPK signaling systems

We showed that different RTKs and their cognate GFs produce distinct population distributions of single-cell ERK states. Our results extend the notion that at least some of this specificity is encoded in the MAPK network. We demonstrated that the EGF, NGF, and FGF2

signaling pathways can sense almost four orders of magnitude of GF concentrations and translate them into robust signaling states that are biologically relevant for fate decisions.

The EGF/NGF systems interpret graded inputs into a variety of signaling states, thanks to the modulation of negative/positive feedbacks by their respective RTKs (Ryu *et al*, 2015). At all concentrations (0.25–250 ng/ml), sustained EGF induces switch-like, adaptive responses with a robust high-amplitude initial peak that narrows as the GF dosage increases (Fig 1D–G). Hence, gradual EGF increase leads to a stronger negative feedback and thus faster robust adaptation, which can protect the RTK signaling against stochastic effects due to heterogeneous expression of signaling components (Birtwistle & Kolch, 2011). Low EGF input produces slow adaptive, almost sustained, ERK activity (Figs 1D–G and EV1) sufficient to differentiate a portion of cell population (Fig 5A and B). High EGF levels lead to low differentiation due to fast adaptation of ERK activity. Thus, an EGF dose response can already produce sufficiently distinct signaling states to induce different fates. In marked contrast, low sustained NGF input induces only low-amplitude responses, but at high NGF input, the responses become sustained and saturate above 2.5 ng/ml NGF (Fig 1D). Pulsed stimulation indicates a mix of adaptive/sustained ERK activity responses (Figs 2B and EV3B), with the population distribution of the latter increasing at high NGF input. This again documents our previous finding of RTK-modulated positive feedback: Increased NGF input leads to stronger positive feedback, consequently increasing the fraction of cells with sustained ERK activation (Ferrell & Machleder, 1998). However, saturation of ERK activity limits the gradual increase of the portion of cells with sustained signaling (Fig 1D–G). The NGF–TrkA–MAPK network varies the distribution of adaptive and sustained responses, and thus can shift the population distribution of proliferation/differentiation fates in response to increasing input, as previously proposed (Chen *et al*, 2012).

In contrast to the EGF/NGF systems, the FGF2–FGFR–MAPK network interprets the input concentration using a system that comprises competing receptor complexes in the extracellular space and a simple intracellular network with a negative feedback. The robustness emerging from this negative feedback might allow FGF2 to reliably integrate a large range of FGF2 concentrations into the gradually changing mix of transient/sustained ERK states. The function of this network might be to enable reliable signal transmission of fate decisions during developmental FGF morphogen gradient interpretation (e.g., FGF2 input might gradually vary the relative abundance of transient/sustained ERK states, evoking different fates along a morphogen gradient; Ornitz & Itoh, 2015).

### Reprogramming fate decisions by dynamic GF application

Our data clearly paint a picture in which the different RTK/MAPK networks decode the binding of their cognate GF by inducing specific population distributions of distinct single-cell ERK states. Understanding how this emerges from network structures provides the attractive possibility to evoke synthetic ERK signaling states of desired duration by simple dynamic GF application. Previously, we had shown that multiple EGF pulses, when delivered at a specific frequency, can induce a synthetic sustained ERK signal that leads to a differentiation fate (Ryu *et al*, 2015). For FGF2, the application of a high dose of sustained versus pulsed FGF2 leads to clear transient versus sustained ERK states (Figs 1D and 2C), which further correlate with absence/presence of differentiation respectively (Fig 5B and D). These results further support the notion that the time ERK remains ON is important for actuation of the differentiation fate (Murphy *et al*, 2002; Gillies *et al*, 2017). However, application of low concentrations of sustained FGF2, while evoking sustained ERK activity (Fig 1D), does not lead to differentiation. This strongly suggests that additional signaling pathways such as the PI3K/AKT pathway are also important for regulation of fate decision (Kim *et al*, 2004; Chen *et al*, 2012), and might require a higher threshold GF concentration.

## Material and Methods

### Reagents and Tools table

| Reagent/resource | Reference or source | Identifier or catalog number |
|---|---|---|
| **Experimental models** | | |
| PC12 EKAR | *Ryu et al* (2015) | |
| PC12 NS1 | Tobias Meyer | |
| **Antibodies** | | |
| Anti-Tubulin DM1A | Sigma | T9026 |
| Alexa 488 anti-mouse | Molecular Probes by Life Technologies | A11029 |
| **Chemicals, enzymes, and other reagents** | | |
| hEGF | Sigma | E9644 |
| NGF-beta human | Sigma | N1408 |
| hBFGF | Sigma | F0291 |
| Dextran Alexa 546 | Thermo Fisher Scientific | D22911 |
| NaClO$_3$ | Sigma | 403016 |
| DAPI | Sigma | D9542 |
| **Software (include version where applicable)** | | |
| CellProfiler | http://cellprofiler.org | V 2.2.1-newest |
| R | https://www.rstudio.com/products/rstudio/download/ | 1.1.453-newest |
| NIS-Elements AR | NIKON | 4.6 |
| Matlab | www.mathworks.com/ | 2017b |
| **Other** | | |
| AnaTag HiLyte Fluor 647 Microscale Protein Labeling Kit | AnaSpec | AS-72050 |

### Methods and Protocols

#### Cell culture

PC-12 cells stably expressing the EKAR2G1 construct, described earlier in Ryu *et al* (2015), and PC-12 Neuroscreen-1 (NS-1, gift from Tobias Meyer) were cultured using low-glucose DMEM (Sigma) supplemented with 10% horse serum (HS; Sigma), 5% fetal bovine serum (FBS; Sigma), and 1% penicillin/streptomycin. Cells were cultured on plastic tissue culture dishes (TPP) coated with 50 μg/ml collagen from bovine skin (Sigma). Cells were passaged at 70% confluence by detaching cells using a cell scraper (Fisher).

#### Microfluidic device fabrication and preparation

Microfluidic device preparation was performed as described previously (Ryu *et al*, 2015). In short, polydimethylsiloxane (PDMS) polymer (Dow Corning) was mixed with the catalyzer in 10:1 ratio in a plastic beaker. A first layer of 4–5 g was poured on the master and then degassed in a desiccator before solidifying at 80°C for 1 h. Eight-well reservoir strips (Evergreen) were divided in 2 and then glued on the first layer using PDMS and solidifying at 80°C for 30 min. Finally, the second layer of 15–20 g of PDMS is used to finalize the device. The PDMS replica was cut and punched at the appropriate inlets and outlets. Plasma treatment was used to bond the

PDMS replica to the 50 × 70 mm coverslip (Matsunami, Japan) to allow proper sealing that resists the high-pressure applied during the experiments. To enhance the bonding strength, the device was heated for 15 min in an 80°C dry oven. After bonding, the device was immediately filled by adding 200 µl of 50 µg/ml collagen solution in PBS to each outlet reservoir and put at 37°C. To increase coating efficiency in the device, 10 µl of the collagen solution was aspirated 3× from the cell seeding port after 1 h each before seeding cells.

PC12/EKAR2G cell suspensions were prepared at a concentration of $10^6$ cells/ml. 50 µl of this cell suspension was added in the outlet and aspirated with a pipette from the cell reservoir inlet port. After a 10′ incubation, residual cells in the outlet were removed by aspiration, and 250 µl of DMEM supplemented with 10% HS, 5% FBS, and 1% penicillin/streptomycin was added to the outlet reservoir. The inlet reservoirs were filled with 150 µl of starving medium (pure low-glucose DMEM). Prior to experiments, outlet reservoirs were emptied and inlet reservoir was filled with 200 µl of fresh starving medium (inlet 1) and starving medium with appropriate GF/NaClO$_3$ concentration and added dextran Alexa 546 (4 nM final concentration) (inlet 2).

### Live cell imaging

All FRET ratio-imaging experiments were performed on an epifluorescence Eclipse Ti inverted fluorescence microscope (Nikon) with a PlanApo air 20× (NA 0.75) objective controlled by NIS-Elements (Nikon). Laser-based autofocus was used throughout the experiments. Image acquisition was performed with an Andor Zyla 4.2 Plus camera at a 16-bit depth. Donor, FRET, and red channel images (to visualize an Alexa 546-dextran that indicates GF exposure) were acquired sequentially using filter wheels. The following excitation, dichroic mirrors, and emission filters (Chroma) were used: donor channel: 430/24×, Q465LP, 480/40 m; FRET channel: 430/24×, Q465LP, 535/30 m; and red channel: ET550/15, 89,000 bs, 605/50 m (for dextran imaging). Standard exposure settings were used throughout the experiments. 440-nm (donor and FRET channel excitation) and 565-nm (red dextran) LED lamps were used as light sources (Lumencor Spectra X light engine), with 3% (440 nm) and 5% (565 nm) of lamp power. Acquisition times were 30 ms for donor channel and 30 ms for FRET at binning 2 × 2 and 100 ms 8 × 8 binning for the red channel. Cells were imaged in DMEM with 1,000 mg/ml glucose, and penicillin/streptomycin, at 37°C. The microfluidic device was mounted on the microscope stage and was connected by the tubing to a CellASIC ONIX (Merck Millipore) pump.

### NS-1 differentiation experiments

A total of 8,000 cells per well (96-well plates; BD Biosciences) were seeded in starving medium consisting of low-glucose DMEM supplemented with 0.2% HS and 1% penicillin/streptomycin. Cells were starved for 24 h before adding 200 µl of the appropriate GF. For 10-min and 3-min experiments, wells were carefully washed once with 200 µl starving medium to dilute residual GF. After 2 days, cells were fixed using 4% PFA at 37°C for 10 min and subsequently washed twice with PBS. Cells were permeabilized in 0.1% Triton X-100 in PBS for 10′ and blocked in PBS with 1% BSA and 22 mg/ml glycine for 30′. Cells were then incubated with an anti-alpha-tubulin (Sigma T9026, 1:1,000) antibody at 4°C overnight, washed 3× for 10′ with PBS, and incubated with an Alexa 546 anti-mouse

secondary antibody (1:1,000) for 3 h. Samples were washed with PBS with DAPI (1:1,000) for 5′ and then washed 2× 10′ with PBS before imaging.

### Image analysis

For the segmentation, tracking, and ratio calculation in time-lapse experiments, we used CellProfiler. First, FRET and donor channels were corrected for uneven illumination using the *CorrectIlluminationCalculate* and *CorrectIlluminationApply* modules using the *Background* setting. Cells were then segmented using the *IdentifyPrimaryObjects* module. As there is no nuclear marker for segmentation, we excluded clumps of cells using stringent size exclusion in this module. We tracked objects using the *TrackObjects* module and calculated the ratio image using the *ImageMath* where the FRET image is divided by the Donor image. Using *MeasureObjectIntensity*, the mean intensity of the newly created ratio image was measured for every tracked object as well as the mean intensity of labeled FGF HiLyte 647 for experiment using it. In addition, total image intensity of the red dextran Alexa 546 channel was measured to follow GF exposure. Measurements were exported as csv files and quantified with R scripts.

We used a different CellProfiler pipeline to analyze differentiation experiments. First, DAPI channel was segmented using *IdentifyPrimaryObjects* to detect nuclei. Using the *IdentifySecondaryObjects* module, cells including their neurites were segmented using the nuclei objects as a seed and the tubulin stain as the image. These objects were then skeletonized using the *ExpandOrShrinkObjects* module. To obtain the soma, a series of morphological operations were applied (4× erode, followed by 4× dilation) to the tubulin images using the "Morph" module; then, the resulting images were segmented again using *IdentifySecondaryObjects*. Total neurite length per cells was then measured using *MeasureNeurons* module, and data were exported to csv files.

### Quantification and statistical analysis
#### Clustering

We used R software to analyze and cluster time series. The amplitude of each trajectory was first normalized to its own mean before GF stimulation, i.e., $t \in [0, 40]$ for Figs 1C and D, 2A–C, 3A,C,E, and 4A and B, or $t \in [36, 40]$ for Figs 1E and 3B,D,F.

For clustering of sustained and single-pulse GF stimulations, we used dynamic time warping from *dtw* R package. The subsequent hierarchical clustering was performed using standard R functions *dist* and *hclust*.

#### PC analysis

We use standard R function *prcomp* for principal component analysis (PCA). For the decomposition, we use pooled data for all GFs (EGF, NGF, and FGF2) and their concentrations (0.25–250 ng/ml) from Fig 1E (main text). After the decomposition, we add negative control dataset (no GF) by rotating it to the new PC basis.

#### Population distance

With this novel approach, we set out to quantify the separation between two populations of single-cell time series as shown in Figure S2B. The two populations may correspond to single-cell dynamic ERK responses to two treatments with different GFs or different GF concentrations.

In the first step, at every measured time point we calculate distance between two distributions of a measured quantity (middle panel of Appendix Fig S2B). We use Jeffries–Matusita distance ($d_{JM}$), which, for two normalized histograms $p$ and $q$, with $\Sigma_i p_i = 1$, reads:

$$d_{p,q}^{JM} = \sum_{i=1}^{n} (\sqrt{p_i} - \sqrt{q_i})^2, d_{p,q}^{JM} \in [0, 2]$$

where $n$ is the number of histogram bins. The J-M distance is bound and equals 0 for two identical distributions, and 2 for two entirely disjointed distributions regardless of how far apart they are. As a result, $d_{JM}$ overemphasizes small and suppresses high separability values.

In the second step, we calculate the fraction of area under the $d_{JM}$ curve over time, relative to maximum AUC of $d_{JM}$. Therefore, the population distance between two sets of single-cell time series, $a$ and $b$, reads:

$$d_{a,b}^{popul} = \frac{\sum_{k=1}^{N} d_{(a,b)_k}^{JM}}{2Ndt}, d_{a,b}^{popul} \in [0, 1]$$

where $N$ is the number of time points, $dt$ is time interval of data, and index $k$ indicates a time point at which $d_{JM}$ is calculated.

### Mechanistic modeling

Appendix Table S1 shows all modeled species, their notation used for the equation, and the initial values. The initial values for the FGF2 receptor, the negative feedback species, and HSPG are inferred from the data and are modeled through the indicated parameters. The other initial values were taken from http://bionumbers.hms.harvard.edu.

The model equations for the basic model are shown in Appendix Table S2. The phosphorylation events are modeled with the Michaelis–Menten kinetics. The negative feedback is modeled through the modeling species *NFB* and its "active" version *NFB*\*, which affects the phosphorylation rate of *RAF* in a Hill-type manner, with Hill coefficient $h_{nfb}$. The receptor models are shown in Appendix Table S3. Since the activation happens through direct binding of FGF2 to HSPG and then to FGFR, these reactions have linear propensities. Even though the activation of FGFR requires dimerization (and possibly multimerization), we modeled it linearly to avoid making the model unnecessarily complicated.

The regulation of the negative feedback is model dependent (equations in Appendix Table S4). For the incoherent feed-forward models (C1, C2, C3), it is only regulated by the membrane species *HFR* and *FGFR*\*; for the negative feedback models (B1, B2, B3), it is only regulated through *ERK*\*; and for the models (D1, D2, D3), it is regulated through both the receptor species *HFR* and *FGFR*\* as well as *ERK*\*. Since the NFB does not correspond directly to any biological species, we modeled its activation as Hill dynamics with inferred Hill coefficients $h_{hfr}$ and $h_{fgfr}$.

Model files are deposited in https://github.com/Mijan/LFNS_MSB/tree/MSB_version/FGF2_models.

### Parameter estimation

All parameters are listed in Appendix Table S5. For the parameter estimation, we used a custom implementation of a NS method based on Skilling (2006). Our custom implementation of NS can be obtained from the GitHub repository (https://github.com/Mijan/LFNS_MSB). This repository also contains all model files, as well as the results from all of our inference runs. Given the experimental data, NS approximates the posterior (being the Bayesian evidence and being the likelihood) by drawing samples of parameter vectors from the prior, along with weights that are used to recover the posterior distribution and the Bayesian evidence. NS explores areas of the prior constrained to higher regions of likelihood corresponding to an increasing sequence of thresholds. For a thorough discussion on the NS approach, see, for instance, Skilling (2006). For termination criteria, we follow Skilling (2006) and ran the NS algorithm until the final prior volume multiplied by the highest likelihood in that volume is smaller than 0.001 times the current evidence estimate and the difference between the highest log-likelihood and the lowest log-likelihood in the current "live" set is < 2. The NS algorithm was run in parallel on 48 cores, and the sampling from the constrained prior in each iteration was by performing density estimation on the current "live" points and using rejection sampling to sample from the prior on the support of the constrained prior. The priors for all parameters were chosen to be log-uniform.

## Data availability

CellProfiler 2.2.1 pipelines to process time-lapse movies and snapshots from differentiation experiments are provided as Code EV1. Source code for the inference algorithm, all model files (in human-readable.txt format), and all results of the inference runs (all parameter posteriors as well as all intermediate algorithm results) are available at https://github.com/Mijan/LFNS_MSB.

**Expanded View** for this article is available online.

## Acknowledgements

This work has been supported by grants from the Swiss National Science Foundation and the Novartis Foundation for medical–biological research to O.P., from the Korean–Swiss Science and Technology Programme to O.P. and N.L.J., from the Basic Science Research Program through the National Research Foundation of Korea (NRF) funded by the Ministry of Science and Technology (NRF 2018R1A2A1A05019550) to N.L.J., and from the European Union's Horizon 2020 research and innovation program under grant agreement No. 730964 (TRANSVAC project) to M.K. We thank Tobias Meyer for the PC-12 NS-1 cell line.

## Author contributions

OP, YB, MD, and HR conceived the study. YB and HR performed experiments. YB, MD, and M-AJ analyzed the data. JM and MK performed mathematical modeling. HR and NLJ set up the microfluidic platform. OP supervised the work. OP, MD, JM, and YB wrote the paper.

## Conflict of interest

The authors declare that they have no conflict of interest.

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
