## [Review Process File · Molecular Systems Biology]

Temporal perturbation of ERK dynamics reveals network architecture of FGF2-MAPK signaling

Yannick Blum, Jan Mikelson, Maciej Dobrzyński, Hyunryul Ryu, Marc-Antoine Jacques, Noo Li Jeon, Mustafa Khammash and Olivier Pertz.

Review timeline:

Submission date:	9 th April 2019
Editorial Decision:	31 st May 2019
Revision received:	27 th September 2019
Accepted:	21 st October 2019

Editor: Maria Polychronidou

Transaction Report:

1st Editorial Decision

31st May 2019

Thank you again for submitting your work to Molecular Systems Biology. We have now heard back from the three referees who agreed to evaluate your study. As you will see below, the reviewers acknowledge that the presented findings seem interesting. They raise however a series of concerns, which we would ask you to address in a revision.

I think that the reviewers' recommendations are rather clear and there is therefore no need to repeat the comments listed below. All three reviewers provide constructive suggestions for improving the study. As you may already know, our editorial policy allows in principle a single round of major revision so it is essential to provide responses to the reviewers' comments that are as complete as possible. Please feel free to contact me in case you would like to discuss in further detail any of the issues raised by the reviewers.

REFeree REPORTS

Reviewer #1:

Blum et al. present a systems analysis of growth factor-induced MAPK signaling and cell fate decisions in PC12 cells. Using live-cell imaging of Erk signaling, they compare EGF, NGF and FGF2 stimulation, and conclude that FGF2 induces complex MAPK activation patterns not seen in response to the other two ligands, including a bell-shaped dose-response, a late response rebound and anti-phasic responses to repeated FGF2 pulses. The authors extensively quantify signaling heterogeneity at the single-cell level using time course clustering and employ mathematical modeling to identify the underlying molecular mechanisms.

The article is clearly written and of high quality. The biological question is of high relevance and of general interest, as PC12 neuronal differentiation serves as a paradigm for cell fate decisions. The

strength of this article mainly lies in the experimental analysis of single-cell signaling dynamics under perturbation conditions. The results provide novel insights into the encoding of cellular decisions based on the temporal dynamics of signaling. That said, the link between signaling dynamics and cell differentiation should be quantitatively analyzed in more detail. Furthermore, several aspects of the model selection framework need clarification as detailed below.

MAJOR POINTS

1) The authors mainly use qualitative arguments to link Erk signaling dynamics to neuronal differentiation outcomes and unspecifically speak about correlation between signaling states and differentiation. They should provide more quantitative measures for the link between signaling dynamics and differentiation.

They also mention that differentiation at 2.5 ng/ml FGF2 does not correlate with the observed MAPK single-cell trajectories. They should quantify this failure of Erk dynamics to predict cellular outcomes (by comparing single-cell responses at 0.25 ng/ml 2.5 ng/ml NGF and FGF2, respectively). They should discuss possible explanations, e.g., differential activation of the PI3K signaling pathway.

2) p12: The model variants of MAPK signaling seem to be chosen to some extent arbitrarily. Why didn't the authors consider a positive feedback model (Santos, 2007)? Does evidence from the literature support the existence of positive/negative feedback, receptor downregulation and/or IFF in the FGF signaling cascade? The final chosen model (B3) fails to fit the late rebound of Erk activity at 250 ng/ml, suggesting that a negative feedback is not the only relevant adaptation mechanism in the system.

3) p12/13: Surprisingly, the authors do not use all previously measured FGF doses and types of pulsatile stimulation (Figs. 2-4) to calibrate and/or validate their models. In particular, it seems questionable to use only two FGF concentrations (2.5 and 250 ng/ml) to constrain the dose-dependent behavior of a signaling system during model fitting. The authors should consider using all conditions for fitting/validation, as this should better constrain the parameter values and predictions. Alternatively, they should compare the existing final model to all available experimental conditions in a Supplemental Figure.

4) p12: The authors mention that 10 out of 12 models faithfully reproduce the experimental data. They should provide a quantitative measure for model selection.

5) p15: Concerning their mechanistic predictions in the Discussion, the authors should briefly describe the affinities and temporal dynamics in the two competing pathways to further clarify the following aspects: Why is a switch from FGFR-HSPG-FGF2 to FGFR-FGF2 observed at increasing FGF2 doses? Why does FGF2 relocate to FGFR-HSPG-FGF2 upon washout at high doses?

MINOR POINTS

1) p5: For comparison, the authors should report the range of FRET biosensor ratios in unstimulated cells

2) p5: The term biphasic Erk activation has also been used to describe temporally repetitive profiles. The authors should more explicitly explain what they mean with biphasic.

3) p6: Clustering with DTW should be explained in more detail. The authors should provide a reference for this type of clustering analysis

4) In Fig. 1F, six time course clusters were identified. By which measure did the authors decide on six clusters?

5) The authors may consider choosing more distinguishable cluster colors in Fig. 1F

6) p12: the authors refer to Birtwistle et al. (2011) concerning the modeling of Erk activity

measurements. They should briefly explain the procedure

Reviewer #2:

This paper extensively looks into ERK activity in response to FGF2 stimulation and compares and distinguishes it from EGF and NGF stimulation. Dynamic ERK activity was obtained and studied for a time of 200 minutes, through live cell imaging based on the biosensor EKAR2G. Further, the ERK activity was linked to differentiation of cells based on neurite outgrowth. The motivation and methods used in the paper are significant. Study on dynamic ERK activity is important because of its implications in cell differentiation and growth and there has not been too much study on FGF2 induced ERK activity, especially extensive dynamic study. The paper finds that ERK activity in response to FGF2 stimulation is distinct from both EGF and NGF stimulation and evokes distinct cell differentiation as well. The ERK activity and linked phenotypes in response to FGF2 is an important experimental result and should be published. The mechanistic modeling associated is also important as it helps to clarify the signaling network structure that controls this system and how it might be different from other ligands. However there are a few concerns as mentioned below.

Major Concerns:

1. The paper claims that MAPK pathways are differentially wired through different RTK systems and the ERK activity is linked to cell differentiation. However past literature suggests that the PI3K-AKT pathway also significantly affects cell differentiation [1]. It is conceivable that the PI3K-AKT pathway has also been differentially wired through FGF2 stimulation and that could be having an effect on the resulting phenotype. A more complete picture of the phenotypic response to FGF2 stimulation should also include AKT activity.
2. ERK activity not only affects differentiation but also cell proliferation as the paper itself mentions multiple times. The effect of FGF2 on cell proliferation could be an important result that could also be related to the kinds of transient and sustained ERK activity obtained through different doses of FGF2, and particularly it is unclear how it affects the intended differentiation phenotype. A discussion on the interaction between the two phenotypes and how it might impact the interpretations in the paper is warranted. Any data already collected along these lines would be a very important addition to the paper as well.
3. The paper claims and shows that NaClO₃ mediated HSPG inhibition, results in decrease of ERK amplitude and rescue of in-phase ERK activity. However the effect of HSPG inhibition and the resultant ERK activity on the cell phenotype (differentiation) is not explored. (It may be possible that the phenotype resulting from ERK activity caused by HSPG inhibition could be distinct from the phenotype resulting from similar ERK activity caused by a dose change of FGF2. Looking at the phenotype with HSPG inhibition could either confirm or negate this idea and may further support/negate the fact that the phenotype is highly dependent on the ERK activity).
4. The rationale and building of clusters is somewhat difficult to understand and the plot is sometimes unclear. The trajectories of clusters with the same color code in 3' pulse, 10' pulse and 60' pulse are all different. Besides they seem to have different population distribution as well. How were the different trajectories and populations decided? Perhaps by considering a slightly different population sample, a different trajectory can be obtained. It seems that one color code across 3' pulse, 10' pulse and 60' pulse is thought to represent a similar ERK activity, but it is sometimes not very clear if that is the case. For instance in figure EV3, in 3' and 10' pulses of FGF2, it is not very clear if the blue cluster and the light pink cluster are representing the similar ERK activity between the two kinds of pulses. Could there be a better and easier way to represent ERK activity cluster? Perhaps by having one trendline for a cluster across all three kinds of pulses and seeing how many cells cluster towards each trendline.

Minor concerns:

1. It is claimed in figure 3, 25 ng/ml FGF2 evoked a roughly 50% population distribution of in- and anti-phase scTEPs- but it seems more like 35%-65% of in- and anti-phase scTEPs.

2. Also in figure 3, it is claimed desensitization occurred at the timescale of hours, but this was not dependent on the NGF concentration as for EGF- but it seems that the concentration of NGF does have some effect-2.5 ng/ml of NGF shows very little desensitization and 25 and 250 ng/ml NGF shows desensitization.

3. Clarify use of bi-phasic in introduction.

1. Kim Y, Seger R, Cv SB, Hwang SY, Yoo YS. A Positive Role of the PI3-K/Akt Signaling Pathway in PC12 Cell Differentiation. *Mol. Cells* 2004;18:353-359.

Reviewer #3:

The paper by Blum et al studies the dynamic of Erk in PC12 cells in response to FGF2 ligand. The work contrasts Erk dynamics with that of two other growth factors (EGF and NGF) to understand better how differences in dynamics effect cell decision. They utilize interesting and innovative model selection approach to provide mechanistic insights on the wiring of receptors and feedback scheme that could plausibly generate the observed dynamics. Overall I found the paper interesting and the work rigorous. I recommend it be accepted for publication in MSB.

A few comments, none of them "major" in the sense that they should preclude publication, but I hope the authors will find these useful and will want to address them.

The authors should make a better distinction in the text between parameter inference and model selection. As far as I understand, the Bayesian aspect of the work was the parameter inference and the comparison between the 12 models was done "by eye". This is a legitimate approach but is just confusing given the large literature on Bayesian model selection that actually compares between models using BIC criterion etc. The authors could add a formal model selection step or simply clarify that the model selection was qualitative and the only "Bayesian" step was parameter inference. Sentences like: "...model selection using a Bayesian parameter inference" are very confusing given what was actually done.

In the model selection, it would have been nice to see that the final feedback structure that was recovered agrees with EGF and NGF data. While the receptor component of the models is different, the feedback structure should be the same. Erk response in PC12 cells has been analyzed in depth and my impression is that there are subtle differences between different model generations (i.e. compare the current model to Santos 2007). Including EGF/NGF in the model and contrasting the results with past models will be very informative to the reader.

The acronyms scTEP and paTEP are unnecessary and make it harder to read.

The introduction could better emphasize the knowledge gap addressed here. As currently written it provides a nice background to GF signaling in PC12 cells but reading it I didn't fully understand what is the problem that this paper will address. Another paragraph or two before the final "Here we explore..." will really help readers.

Given the emphasis on single cell dynamic data in the paper, it felt like a missed opportunity that the authors did all the model fitting on population averages. Given that the single cells data was classified into "clusters" (Figures 1+3) it would make more sense to not average all cells before the fit. Authors could either fit but cluster average or go all the way and do single cell parameter fitting (see Yao et al MSB 2017).

To what degree are other non-Erk dynamic factors contribute to the proliferation/differentiation decision? Chen et al (PMID: 22206868) showed that Akt plays a key role as well. The interpretation of Figure 5 in the text implies that Erk dynamics differences explain everything. Not ignoring Akt activity will be important.

Reviewer #1:

Blum et al. present a systems analysis of growth factor-induced MAPK signaling and cell fate decisions in PC12 cells. Using live-cell imaging of Erk signaling, they compare EGF, NGF and FGF2 stimulation, and conclude that FGF2 induces complex MAPK activation patterns not seen in response to the other two ligands, including a bell-shaped dose-response, a late response rebound and anti-phasic responses to repeated FGF2 pulses. The authors extensively quantify signaling heterogeneity at the single-cell level using time course clustering and employ mathematical modeling to identify the underlying molecular mechanisms.

The article is clearly written and of high quality. The biological question is of high relevance and of general interest, as PC12 neuronal differentiation serves as a paradigm for cell fate decisions. The strength of this article mainly lies in the experimental analysis of single-cell signaling dynamics under perturbation conditions. The results provide novel insights into the encoding of cellular decisions based on the temporal dynamics of signaling. That said, the link between signaling dynamics and cell differentiation should be quantitatively analyzed in more detail. Furthermore, several aspects of the model selection framework need clarification as detailed below.

MAJOR POINTS

1) The authors mainly use qualitative arguments to link Erk signaling dynamics to neuronal differentiation outcomes and unspecifically speak about correlation between signaling states and differentiation. They should provide more quantitative measures for the link between signaling dynamics and differentiation.

Our experimental setup allows only for an approximate quantification and qualitative statements about the relationship between signaling dynamics and cell fate. ERK responses to GF stimulations are measured on the timescale of hours, while differentiation is measured in a different experiment on the timescale of 2 days. Thus, any relationship drawn from these two experiments has to be performed on population averages, since we cannot track ERK dynamics and fate in the same single cells because of limitations of the microfluidics setup, and of our photon budget. For the reviewer's convenience, we now have performed such a quantification. For every treatment shown in Figure 5 (sustained and 3' GF stimulation), we have correlated the mean accumulated ERK activity over time (area under ERK activity curve after the 1st peak), with the differentiation ratio. In this correlation, two data points (0.25 ng/ml FGF2 sustained, 250 ng/ml NGF 3' pulse) are clearly outliers. The Pearson product moment correlation coefficient is +0.90 (95% CI = [0.77, 0.96]) when these data points are omitted, and +0.75 (95% CI = [0.51, 0.89]) when they are retained. This shows that there is a good correlation between integrated ERK activity over time and the differentiation fate, which is indeed more robust when the two outliers are discarded.

We believe that such correlation plots, while instructive, should not be included in the manuscript. The strong points that we want to make are: 1. that there is a biphasic response in ERK activation dynamics in response to increasing dosage of sustained FGF; 2. that a pulse of high FGF concentration, that evokes a 1st ERK activity peak followed by a sustained rebound of ERK activity leads to stronger differentiation (that is absent upon stimulation with high sustained FGF) leads to potent differentiation.

We believe that the outliers originate because of the following reasons:

0.25 ng/ml sustained FGF2 leads to sustained ERK activity (Fig.1D), but not result in a clear differentiation fate. We previously observed a similar conflicting result when a multi-pulse NGF stimulation regime optimized to yield sustained ERK activity is applied (Ryu et al., 2015). Here, sustained ERK activity profiles that look identical when low or high NGF concentrations are applied, only led to potent differentiation with high NGF doses. We attribute this failure to limited signaling strength of additional pathways such as PI3K/AKT signaling (Chen et al., 2012; Kim et al., 2004) that might also feed into differentiation but fail to be activated at such a low FGF2 concentration. We have now included this in the Discussion section (p.20, paragraph 3).

250 ng/ml 3' NGF pulse leads to robust differentiation, but induces less ERK activity under the curve compared to sustained NGF stimulation. We can image a scenario in which other signaling pathways (again PI3K/AKT signaling, for example) can compensate for the lower ERK activity. We have not discussed this further.

They also mention that differentiation at 2.5 ng/ml FGF2 does not correlate with the observed MAPK single-cell trajectories. They should quantify this failure of Erk dynamics to predict cellular outcomes (by comparing single-cell responses at 0.25 ng/ml 2.5 ng/ml NGF and FGF2, respectively). They should discuss possible explanations, e.g., differential activation of the PI3K signaling pathway.

We assume that reviewer #1 means that 0.25 ng/ml FGF2 ERK dynamics does not correlate with differentiation. The 2.5, 25, and 250 ng/ml FGF2 are correlating with differentiation. We have addressed these concerns in the precedent section.

2) p12: The model variants of MAPK signaling seem to be chosen to some extent arbitrarily. Why didn't the authors consider a positive feedback model (Santos, 2007)? Does evidence from the literature support the existence of positive/negative feedback, receptor downregulation and/or IFF in the FGF signaling cascade? The final chosen model (B3) fails to fit the late rebound of Erk activity at 250 ng/ml, suggesting that a negative feedback is not the only relevant adaptation mechanism in the system.

There is indeed much less information about MAPK network wiring known for FGF than for EGF/NGF, where the transient (EGF) and sustained (NGF) ERK activity profiles immediately suggested the presence of negative/positive feedback loops. Previous analyses of FGF signaling displayed mostly transient ERK activity responses. The biphasic dose response of ERK activity and fate outputs in response to FGF2 was previously modeled by Kanodia (Kanodia et al., 2014). This includes a network with multiple extracellular FGFR/HSPG receptor species, and an intracellular MAPK network that involves a negative feedback loop (Kanodia et al., 2014). In our previous publication (Ryu et al., 2015), in which we had used pulsed GF stimulation to dissect EGF/NGF-dependent ERK activity responses, we had shown that receptor downregulation could not account for the observed ERK dynamics. We also had identified an incoherent feed forward (IFF) structure in the network in which the strength of (EGF-dependent) negative and the strength of (NGF-dependent) positive feedback loop are modulated by RTK activity. We therefore reasoned that such an IFF network structure might be a conserved feature for different RTKs. Indeed, our model selection procedure that consists of 1. Bayesian parameter inference using different postulated networks on select training datasets; and 2. Validation of model predictions using alternate datasets; yielded a model that can qualitatively explain the measured ERK activity responses. Indeed, we cannot formally exclude that different MAPK network topologies can explain our datasets, especially since we are already using reductionist MAPK model formulations. The combination of topologies to test would be overwhelming, and beyond the scope of this study. We assume that the reviewer enquires about the possibility of a positive feedback loop because 2.5 ng/ml FGF leads to a sustained ERK activity, as is usually associated with NGF stimulation. To explore if a positive feedback loop can explain such a behavior, we tested a number of models. First, we tested the three extracellular receptor interaction models with an intracellular topology that consists of a positive feedback (Figure S2: Models E1/E2/E3). Second, we tested variants of models B1, C1, D1 that also include the positive feedback in addition to a negative feedback (B1'), an IFF (C1') or a combination of both (D1'). The rationale behind models B1', C1', D1' is to explore if a simple growth factor – FGFR/HSPG interaction scheme without competing FGF2-FGFR interactions, coupled to a positive and/or negative/IFF topologies can explain our data. This would mean that only the intracellular MAPK network topologies are important for specification of dynamic ERK activity states. For models E1 and B1', we clearly observe that a parameter space that can reproduce the training datasets cannot be identified. The remaining models (E2, E3, C1', C2') recapitulate the training sets better than models E1/B1', although still not perfectly. However, these models fail at predicting the validation datasets. This evidence suggest that model B3 is best at describing the FGF2/MAPK network. We have now added text to the paper that mentions these additional modelling experiments presented in Appendix Figure S2 (p16 paragraph 1).

Regarding B3 not being able to reproduce the slow ERK recovery after sustained stimulation. Indeed, the model B3 does not reproduce the slow ERK activity rebound. Instead, the model response shows a much quicker ERK recovery (within minutes) after which ERK activity levels off. Even if this simulated ERK response does not completely capture the data, our validation experiments with additional pulse stimulation schemes (Fig. EV5), and NaClO₃ perturbation (Fig.7A) suggest that B3 is the most faithful at reproducing all datasets.

3) p12/13: Surprisingly, the authors do not use all previously measured FGF doses and types of pulsatile stimulation (Figs. 2-4) to calibrate and/or validate their models. In particular, it seems questionable to use only two FGF concentrations (2.5 and 250 ng/ml) to constrain the dose-dependent behavior of a signaling system during model fitting. The authors should consider using all conditions for fitting/validation, as this should better constrain the parameter values and predictions. Alternatively, they should compare the existing final model to all available experimental conditions in a Supplemental Figure.

There are several reasons why we chose specific datasets for training.

1. ERK activity responses to GF stimulations are heterogeneous. We train the nested sampling approach on population-averaged signaling states, therefore we choose the treatments where the population average best reflects the single-cell population responses. For example, sustained stimulation with 0.25 ng/ml FGF2 is at the lower end of the dose response curve and induces heterogeneous population responses (e.g. some cells display a first ERK activity peak, while others do not, as illustrated in Fig. EV1A and D). Also, for the multipulse stimulation schemes, 2.5 ng/ml

FGF2 leads to a population homogeneous state, in which ERK response are in-phase with FGF2 pulses. The 250 ng/ml FGF2 also leads to a population homogeneous state, in which ERK response are anti-phasic to FGF2 pulses (Figure 3E,F). In contrast, multipulse 25 ng/ml FGF2 stimulation leads to a mix of in-phase and anti-phase ERK responses. Thus, the population average does not reflect the single-cell responses, and cannot be taken into account for training. We therefore cannot take into account all the datasets, neither for fitting validation, nor for comparison of all available experimental conditions. We now have better emphasized this point on p.14 paragraph 3.

2. We do not want to provide redundant information to train the parameter space, rather we want to exploit the minimal feature set in our datasets that is explanatory about FGF2-MAPK signaling. We reasoned that providing the same information multiple times would bias the parameter inference. For example, the single-pulse FGF2 ERK responses are contained in the first response to multiple FGF stimulation. We believe that the 4 signaling states that we have chosen for training represent the essence of FGF2-MAPK signaling features (i.e. sustained 2.5 ng/ml FGF2: transient ERK activity peak with low adaptation followed by a slow rebound in ERK activity; 250 ng/ml FGF2: transient ERK activity peak with robust adaptation followed by a slow rebound in ERK activity; 2.5 ng/ml multipulse FGF2 stimulation: ERK activity in phase with pulse pattern; 250 ng/ml multipulse FGF2 stimulation: ERK activity anti-phase to pulse pattern).

3. An additional reason for not using all available datasets for model fitting is to avoid strong constraints on the models. The nested sampling approach used in this paper infers parameters to fit experimental data quantitatively as determined by the likelihood function. This may result in inferred parameters that have the highest likelihood with regard to the experimental data, but do not reproduce the qualitative response of this data. For example, a curve that has a similar shape as experimental data but is shifted in time or has an amplitude offset.

4) p12: The authors mention that 10 out of 12 models faithfully reproduce the experimental data. They should provide a quantitative measure for model selection.

We have used the Bayesian evidence for model fitting, however we failed to mention this in the main text. We now have modified the manuscript to clearly mention the Bayesian evidence (p14, paragraph 4). For the training dataset, we see that the two discarded models have a Bayesian evidence of roughly 280 orders of magnitude (~670) lower than the other models (~950). Further, we selected the models based on their capability to reproduce qualitative features of our data. These features include: Amplitude of the initial peak after FGF stimulation, post-peak adaptation strength, and slow rebound in sustained stimulation experiments. Presence/absence of phase shift of the ERK response in the multipulse stimulation experiments.

5) p15: Concerning their mechanistic predictions in the Discussion, the authors should briefly describe the affinities and temporal dynamics in the two competing pathways to further clarify the following aspects: Why is a switch from FGFR-HSPG-FGF2 to FGFR-FGF2 observed at increasing FGF2 doses? Why does FGF2 relocate to FGFR-HSPG-FGF2 upon washout at high doses?

With respect to question 1, we are not sure which switch is meant by the reviewer but we assume that the question concerns the decrease of the trimeric complex upon the second pulse stimulation with high concentration of FGF2. This decrease is due to the fact that upon sudden increase of FGF2 concentration the corresponding reaction flux changes suddenly. At high concentrations of FGF2, the binding of FGFR to FGF2 happens much faster than the binding of HSPG-FGF2 to FGFR, resulting in a suddenly decreasing concentration of the trimeric complex in favor of the dimeric FGFR-FGF2 complex. With respect to question 2, the interaction of FGF2 to HSPG is stronger than to FGFR, thus FGF2 trapped in the HSPG glyocalyx will lead to FGFR-HSPG-FGF2 complexes that are capable of strong signaling. We now have spelled this out more strongly in the discussion (p18 end of paragraph 2).

MINOR POINTS

1) p5: For comparison, the authors should report the range of FRET biosensor ratios in unstimulated cells

In Figure EV1A, we have added a box-plot of single-cell ERK activities measured at 10 minutes before and after sustained GF stimulation. (p7, paragraph 1)

2) p5: The term *biphasic Erk activation* has also been used to describe temporally repetitive profiles. The authors should more explicitly explain what they mean with *biphasic*.

By *biphasic*, we mean a biphasic dose response as in [Kanodia 2014], which means that the ERK activity at a specific time point displays a biphasic behavior. This does not refer to a dynamic ERK behavior. We have now written out “biphasic dose response” in each instance where “biphasic” has been used. We have clarified in the introduction this concept of biphasic dose response (page 5, paragraph 1).

3) p6: Clustering with DTW should be explained in more detail. The authors should provide a reference for this type of clustering analysis

We added a reference in the main text and added a brief explanation (page 7, paragraph 2).

4) In Fig. 1F, six time course clusters were identified. By which measure did the authors decide on six clusters?

We empirically explored different cuts of the dendrogram to highlight major dynamic patterns. After a couple of iterations, we decided to highlight 6 dendrogram branches that correspond to 6 major dynamic patterns. We modified Fig 1F to include the dendrogram.

5) The authors may consider choosing more distinguishable cluster colors in Fig. 1F

We modified the color palette in Fig. 1E-G. This color-palette was originally chosen to color-blind friendly.

6) p12: the authors refer to Birtwistle et al. (2011) concerning the modeling of Erk activity measurements. They should briefly explain the procedure

We added a short explanation in the text (p14, paragraph 2).

Reviewer #2:

This paper extensively looks into ERK activity in response to FGF2 stimulation and compares and distinguishes it from EGF and NGF stimulation. Dynamic ERK activity was obtained and studied for a time of 200 minutes, through live cell imaging based on the biosensor EKAR2G. Further, the ERK activity was linked to differentiation of cells based on neurite outgrowth. The motivation and methods used in the paper are significant. Study on dynamic ERK activity is important because of its implications in cell differentiation and growth and there has not been too much study on FGF2 induced ERK activity, especially extensive dynamic study. The paper finds that ERK activity in response to FGF2 stimulation is distinct from both EGF and NGF stimulation and evokes distinct cell differentiation as well. The ERK activity and linked phenotypes in response to FGF2 is an important experimental result and should be published. The mechanistic modeling associated is also important as it helps to clarify the signaling network structure that controls this system and how it might be different from other ligands. However there are a few concerns as mentioned below.

Major Concerns:

1. The paper claims that MAPK pathways are differentially wired through different RTK systems and the ERK activity is linked to cell differentiation. However past literature suggests that the PI3K-AKT pathway also significantly affects cell differentiation [1]. It is conceivable that the PI3K-AKT pathway has also been differentially wired through FGF2 stimulation and that could be having an

effect on the resulting phenotype. A more complete picture of the phenotypic response to FGF2 stimulation should also include AKT activity.

We completely agree with the reviewer on this point. PI3K-AKT signaling most certainly also is involved in fine tuning of fate decision. It is clear that synthetic ERK activity states optimized for long ERK_{ON} times, that we can engineer by dynamically applying GFs (this and our previous study (ryu et al)), do not always lead to robust differentiation fates. As reviewer 1 correctly states, this might be explained by contributions of PI3K/AKT in addition to MAPK/ERK signaling. We have amended the text to convey this concept, and cited the reference suggested by the reviewer (p 20, paragraph 3)

It will be a significant task to systematically map the contributions of PI3K/AKT signaling networks to fate decisions, which will require understanding different scales. PIP3-produced by PI3K might be very important for neuronal outgrowth by locally controlling cytoskeletal dynamics in the growth cone, for example through Rac1 activation. This signaling will fluctuate on a much faster time (seconds) and a shorter length (single microns) scales than ERK signaling. Regarding AKT signaling, an important paper of Tobias Meyer's lab clearly shows that an ERK/AKT signaling code is important for fate decisions (Chen et al., 2012). However, this signaling code is measured at 24 hours after NGF stimulation, and acute ERK/AKT signaling responses are not indicative of fate decisions. This emphasizes the spatio-temporal complexity of these signaling pathways, with different contributions at different times and locations.

2. ERK activity not only affects differentiation but also cell proliferation as the paper itself mentions multiple times. The effect of FGF2 on cell proliferation could be an important result that could also be related to the kinds of transient and sustained ERK activity obtained through different doses of FGF2, and particularly it is unclear how it affects the intended differentiation phenotype. A discussion on the interaction between the two phenotypes and how it might impact the interpretations in the paper is warranted. Any data already collected along these lines would be a very important addition to the paper as well.

We looked at proliferation in our experimental conditions. In the long-term differentiation experiments, we always include 0.3% horse serum (HS) to warrant cell survival. This is not required in the microfluidics experiments where the experiments only last 6 hours. Thus, in the GF-induced differentiation experiments, we starve the cells for 24 hours in presence of 0.3% HS, then we add GFs in presence of 0.3% HS for 48h). We use these low serum concentrations to stay as close as possible to the experimental conditions used in our biosensor assays (no serum). In this experimental setup, we have not seen major proliferation and also no major differences between the different GFs (EGF/NGF/FGF at different concentrations (determined by cell count 1h and 48h after GF addition). Thus, we have difficulties to induce the proliferation fate with these experimental conditions. We suspect that other labs have fine-tuned experimental conditions (serum concentration, coating, seeding density, ...) to promote the proliferation fate. This is in-line with previous results where EGF-induced proliferation in PC-12 cells required the presence of higher serum concentration (Chen et. al). This is why we specifically focused on the differentiation fate, and did not extensively explore the proliferation fate. We believe that this does not affect the interpretation of our data.

3. The paper claims and shows that NaClO3 mediated HSPG inhibition, results in decrease of ERK amplitude and rescue of in-phase ERK activity. However, the effect of HSPG inhibition and the resultant ERK activity on the cell phenotype (differentiation) is not explored. (It may be possible that the phenotype resulting from ERK activity caused by HSPG inhibition could be distinct from the phenotype resulting from similar ERK activity caused by a dose change of FGF2. Looking at the phenotype with HSPG inhibition could either confirm or negate this idea and may further support/negate the fact that the phenotype is highly dependent on the ERK activity).

We have tried such differentiation experiments. We have the problem that NaClO₃-treated cells are much less adhesive than non-treated cells, and immediately detach when we apply GFs to them. This very likely occurs because NaClO₃ impairs sulfation of multiple extracellular proteins such as ECM proteins, cell adhesion molecules (such as NCAM, cadherins, ...), which are important for neuronal outgrowth in addition to HSPGs. The inability of cells to properly adhere to their substrate because of this perturbation likely also impairs neuronal outgrowth, and provide a confounding factor in these experiments that does not warrant a simple interpretation of the differentiation status.

4. *The rationale and building of clusters is somewhat difficult to understand and the plot is sometimes unclear. The trajectories of clusters with the same color code in 3' pulse, 10' pulse and 60' pulse are all different. Besides they seem to have different population distribution as well. How were the different trajectories and populations decided? Perhaps by considering a slightly different population sample, a different trajectory can be obtained. It seems that one color code across 3' pulse, 10' pulse and 60' pulse is thought to represent a similar ERK activity, but it is sometimes not very clear if that is the case. For instance, in figure EV3, in 3' and 10' pulses of FGF2, it is not very clear if the blue cluster and the light pink cluster are representing the similar ERK activity between the two kinds of pulses. Could there be a better and easier way to represent ERK activity cluster? Perhaps by having one trendline for a cluster across all three kinds of pulses and seeing how many cells cluster towards each trendline.*

In figure 1, we pooled all the sustained GF stimulation and performed hierarchical clustering to identify general ERK dynamic patterns that differentiate EGF, NGF and FGF2. In all remaining figures (Fig. EV1C, EV3 and 3), hierarchical clustering is performed separately for each GF treatment/stimulation pattern. This is simply because clustering would not be able to separate subtle dynamic signaling states if all the experimental conditions were pooled together. For example, in the first panel of Figure EV3A, we clustered pooled data from 3 concentrations of EGF stimulation with a 3' pulse. Even though the color palette is shared across the figure, the patterns indicated with the same color in different panels are unrelated. We thank the reviewer for pointing this confusing issue and we have clarified it in the figure caption (Figure EV3 caption, p35). The dendrogram threshold that decide how many clusters are identified involves visual evaluation of the quality of the clusters. Note that as requested by reviewer 1, we now also provide the dendrogram tree for the clustering of the sustained responses (Fig.1E).

Minor concerns:

1. *It is claimed in figure 3, 25 ng/ml FGF2 evoked a roughly 50% population distribution of in- and anti-phase scTEPs- but it seems more like 35%-65% of in- and anti-phase scTEPs.*

We have now changed the text to mention the roughly 35-65% population distribution (page 10, end of paragraph 2).

2. *Also in figure 3, it is claimed desensitization occurred at the timescale of hours, but this was not dependent on the NGF concentration as for EGF- but it seems that the concentration of NGF does have some effect-2.5 ng/ml of NGF shows very little desensitization and 25 and 250 ng/ml NGF shows desensitization.*

At 2.5 ng/ml, there is lower NGF pulse peak amplitude (Fig.3B). This happens because the low NGF pulse concentration does not allow a full amplitude ERK activity peak as observed at higher NGF pulse concentrations. The hour-scale desensitization is less evident simply because the peak amplitude is lower. It is however clear in the cluster means (Fig.3D) that there is some desensitization, even at low 2.5 ng/ml NGF, compared to high concentrations. In our previous publication (Ryu et al., 2015), we already had compared 5 ng/ml and 50 ng/ml NGF concentrations in multipulse challenges, and had shown and quantified that hour-scale desensitization occurs.

3. *Clarify use of bi-phasic in introduction.*

As already requested by reviewer 1, we have clarified the explanation and written out biphasic dose responses in the main text (p5, paragraph 1).

1. Kim Y, Seger R, Cv SB, Hwang SY, Yoo YS. A Positive Role of the PI3-K/Akt Signaling Pathway in PC12 Cell Differentiation. *Mol. Cells* 2004;18:353-359.

Reviewer #3:

The paper by Blum et al studies the dynamic of Erk in PC12 cells in response to FGF2 ligand. The work contrasts Erk dynamics with that of two other growth factors (EGF and NGF) to understand better how differences in dynamics effect cell decision. They utilize interesting and innovative model

selection approach to provide mechanistic insights on the wiring of receptors and feedback scheme that could plausibly generate the observed dynamics. Overall I found the paper interesting and the work rigorous. I recommend it be accepted for publication in MSB.

A few comments, none of them "major" in the sense that they should preclude publication, but I hope the authors will find these useful and will want to address them.

The authors should make a better distinction in the text between parameter inference and model selection. As far as I understand, the Bayesian aspect of the work was the parameter inference and the comparison between the 12 models was done "by eye". This is a legitimate approach but is just confusing given the large literature on Bayesian model selection that actually compares between models using BIC criterion etc. The authors could add a formal model selection step or simply clarify that the model selection was qualitative and the only "Bayesian" step was parameter inference. Sentences like: "...model selection using a Bayesian parameter inference" are very confusing given what was actually done.

We appreciate the suggestions of the reviewer, and have now precisely mentioned which approaches have been taken at specific steps of our model selection procedure (page 15).

In the model selection, it would have been nice to see that the final feedback structure that was recovered agrees with EGF and NGF data. While the receptor component of the models is different, the feedback structure should be the same. Erk response in PC12 cells has been analyzed in depth and my impression is that there are subtle differences between different model generations (i.e. compare the current model to Santos 2007). Including EGF/NGF in the model and contrasting the results with past models will be very informative to the reader.

The direct comparison between the recovered model and EGF and NGF models is not possible since the inferred parameter for the various feedback loops and receptor dynamics have been inferred with FGF2-specific data and cannot be expected to hold for EGF and NGF-dependent MAPK networks. In order to compare EGF, NGF and FGF models, a completely new inference run with additional data would be necessary. The number of possible model combinations would be huge, and we believe that this would dilute the focus of the paper.

The acronyms scTEP and paTEP are unnecessary and make it harder to read.

We replaced the acronyms with full text.

The introduction could better emphasize the knowledge gap addressed here. As currently written it provides a nice background to GF signaling in PC12 cells but reading it I didn't fully understand what is the problem that this paper will address. Another paragraph or two before the final "Here we explore..." will really help readers.

We have updated the introduction to more strongly emphasize the knowledge gap about FGF2 versus EGF, NGF. We also have provided some strong statements to directly prime the reader to the general relevance of our data to the understanding of RTK/MAPK signaling (p5 end of paragraph 1 and beginning and end of paragraph 2, p6 paragraph 1).

Given the emphasis on single cell dynamic data in the paper, it felt like a missed opportunity that the authors did all the model fitting on population averages. Given that the single cells data was classified into "clusters" (Figures 1+3) it would make more sense to not average all cells before the fit. Authors could either fit but cluster average or go all the way and do single cell parameter fitting (see Yao et al MSB 2017).

We apologize, but could not find the Yao et al MSB 2017 paper). The modelling of the heterogeneity within cell population is an interesting, though challenging topic. First, we would need to determine how to model the heterogeneity, for example by heterogeneous initial concentrations of the involved proteins or by the introducing hyper-parameters on all or some of the model reactions. Further a suitable inference method would need to be selected to be applied to weighted clusters of trajectories. While this investigation of the origin of cell heterogeneity is very

interesting, it lies outside of the scope of the current study that aims at providing a mechanistic explanation for the curious ERK response to FGF2 stimulation.

Our main rationale behind clustering of single-cell ERK responses is to make sure that the resulting population mean responses are not artifacts of averaging a large number of trajectories. This is especially visible in case of the 3-20 FGF2 multi-pulse stimulation. The population mean for the intermediate 25 ng/ml concentration is flat but clustering revealed that the responses consists of a mixture of in- and anti-phasic trajectories. For model fitting we therefore used 2.5 and 250 ng/ml, where the population mean much better reflects the behavior even though single-cell responses are heterogeneous. (p14, paragraph 3)

To what degree are other non-Erk dynamic factors contribute to the proliferation/differentiation decision? Chen et al (PMID: 22206868) showed that Akt plays a key role as well. The interpretation of Figure 5 in the text implies that Erk dynamics differences explain everything. Not ignoring Akt activity will be important.

We indeed agree with the reviewer that PI3K-AKT signaling also is involved in fine tuning of fate decision (as has been mentioned by reviewer 2). It is clear that synthetic ERK activity states optimized for long ERK_{ON} times, that we can engineer by dynamically applying GFs (this and our previous study (Ryu et al., 2015)), do not always lead to robust differentiation fates. As reviewer 1 also correctly states, this might be explained by contributions of PI3K/AKT in addition to MAPK/ERK signaling. We have amended the discussion to convey this concept, and cited the reference suggested by the reviewer. (p 20, paragraph 3)

It will be a significant task to systematically map the contributions of PI3K/AKT signaling networks to fate decisions, which will require understanding different scales. PIP3-produced by PI3K might be very important for neuronal outgrowth by locally controlling cytoskeletal dynamics in the neuronal growth cone, for example through Rac1 activation. This signaling will fluctuate on a much faster time (seconds) and a shorter length (single microns) scales than ERK signaling. The important paper of Tobias Meyer's lab clearly shows that an ERK/AKT signaling code is important for fate decisions (Chen et al., 2012). However, this signaling code is measured at 24 hours after NGF stimulation, at steady-state. Short-term, GF-induced ERK/AKT signaling responses are not indicative of fate decisions in this system. This emphasizes the spatio-temporal complexity of these signaling pathways, with different contributions at different times and locations.

References

- Chen, J.-Y., Lin, J.-R., Cimprich, K.A., and Meyer, T. (2012). A two-dimensional ERK-AKT signaling code for an NGF-triggered cell-fate decision. *Mol. Cell* *45*, 196–209.
- Kanodia, J., Chai, D., Vollmer, J., Kim, J., Raue, A., Finn, G., and Schoeberl, B. (2014). Deciphering the mechanism behind Fibroblast Growth Factor (FGF) induced biphasic signal-response profiles. *Cell Commun Signal* *12*, 34–18.
- Ryu, H., Chung, M., Dobrzyński, M., Fey, D., Blum, Y., Lee, S.S., Peter, M., Kholodenko, B.N., Jeon, N.L., and Pertz, O. (2015). Frequency modulation of ERK activation dynamics rewires cell fate. *Mol. Syst. Biol.* *11*, 838–838.

Thank you again for sending us your revised manuscript. We have now heard back from the two reviewers who were asked to evaluate your study. As you will see below, the reviewers are satisfied with the modifications made and are supportive of publication. I am therefore pleased to inform you that your paper has been accepted for publication in *Molecular Systems Biology*.

REFEREE REPORTS

Reviewer #1:

The authors addressed my comments satisfactorily. Great work!

Reviewer #2:

The authors have done an adequate job of responding to the concerns upon the first round of review and I recommend this paper for publication.

Corresponding Author Name: Olivier Pertz

Manuscript Number: MSB-19-8947